# Dynamic Algorithm for Explainable $k$-medians Clustering under $\ell_p$ Norm

**Konstantin Makarychev**[*]
Northwestern

**Ilias Papanikolaou**[*]
Northwestern

**Liren Shan**[*]
TTIC

## Abstract

We study the problem of explainable $k$-medians clustering introduced by Dasgupta, Frost, Moshkovitz, and Rashtchian (2020). In this problem, the goal is to construct a threshold decision tree that partitions data into $k$ clusters while minimizing the $k$-medians objective. These trees are interpretable because each internal node makes a simple decision by thresholding a single feature, allowing users to trace and understand how each point is assigned to a cluster.

We present the first algorithm for explainable $k$-medians under $\ell_p$ norm for every finite $p \geq 1$. Our algorithm achieves an $\tilde{O}\big(p(\log k)^{1+1/p-1/p^2}\big)$ approximation to the optimal $k$-medians cost for any $p \geq 1$. Previously, algorithms were known only for $p = 1$ and $p = 2$. For $p = 2$, our algorithm improves upon the existing bound of $\tilde{O}(\log^{3/2} k)$, and for $p = 1$, it matches the tight bound of $\log k + O(1)$ up to a multiplicative $O(\log \log k)$ factor.

We show how to implement our algorithm in a dynamic setting. The dynamic algorithm maintains an explainable clustering under a sequence of insertions and deletions, with amortized update time $O(d \log^3 k)$ and $O(\log k)$ recourse, making it suitable for large-scale and evolving datasets.

## 1 Introduction

Artificial intelligence systems play an increasingly important role in everyday life, influencing decisions that affect individuals, businesses, and society as a whole. As their impact grows, so does the need for transparency and human oversight. In response, there is a growing emphasis on making AI decisions understandable to people. This has led to the development of models that aim to present their decision-making processes in a clear and interpretable manner.

In this paper, we study algorithms for explainable clustering. The notion of explainable $k$-means and $k$-medians clustering was introduced by Dasgupta, Frost, Moshkovitz, and Rashtchian (2020) as a way to make clustering decisions more accessible to humans. Both $k$-means and $k$-medians are classical clustering objectives widely used in practice. Here, we focus on $k$-medians clustering under the $\ell_p$ norm. A $k$-medians clustering of a dataset $X \subset \mathbb{R}^d$ is defined by a collection of $k$ centers $c^1, c^2, \ldots, c^k$. Each point $x \in X$ is assigned to the closest center in the $\ell_p$ norm, that is, the center minimizing $\|x - c^i\|_p$. Consequently, every clustering corresponds to a Voronoi partition under the $\ell_p$ norm. The cost of the clustering is defined as

$$\text{cost}_p(X; c^1, \ldots, c^k) = \sum_{i=1}^{k} \sum_{x \in P_i} \|x - c^i\|_p,$$

where $P_i$ denotes the set of points assigned to center $c^i$. We refer to this as *unconstrained $k$-medians clustering*.

---

[*]Equal contribution.

39th Conference on Neural Information Processing Systems (NeurIPS 2025).

While this objective is simple to define and machines can easily compute the nearest centers, the resulting cluster assignments are often difficult for humans to interpret. To make clustering more comprehensible to humans, Dasgupta et al. (2020) proposed using threshold decision trees to represent clusterings. They referred to this approach as *explainable* $k$-means and $k$-medians. For $k$-medians, they considered the $\ell_1$ norm. In a threshold decision tree, each internal node compares a single coordinate of the input to a threshold and directs the point to the left or right subtree accordingly. Each leaf of the tree represents a cluster. We denote the center assigned to $x$ by the decision tree as $\mathcal{T}(x)$. The cost of the clustering is then defined similarly to the unconstrained case:

$$\text{cost}_p(X, \mathcal{T}) = \sum_{x \in X} \|x - \mathcal{T}(x)\|_p.$$

Assigning a data point to a cluster using a threshold decision tree avoids complex distance computations and instead follows a simple, transparent process: each decision is based on a sequence of threshold comparisons. This makes it clear how a particular assignment was made and which features influenced it.

The central question is how much clustering quality is lost in exchange for interpretability. This trade-off is captured by the *the cost of explainability* or *competitive ratio*, defined as the worst-case ratio between the cost of the explainable clustering and that of the optimal unconstrained $k$-medians clustering:

$$\max_X \frac{\text{cost}_p(X, \mathcal{T})}{\text{OPT}_{k,p}(X)},$$

where $\text{OPT}_{k,p}(X) = \min_{c_1,\ldots,c_k} \text{cost}_p(X; c_1, \ldots, c_k)$ denotes the cost of the optimal (unconstrained) $k$-medians clustering of $X$.

Dasgupta et al. (2020) showed—perhaps surprisingly—that the competitive ratio for explainable $k$-medians under the $\ell_1$ norm does not depend on the number of points in the dataset and can be bounded solely as a function of $k$; specifically, it is at most $O(k)$. They also established a lower bound of $\Omega(\log k)$. This result sparked significant interest and led to extensive study of explainable $k$-medians under the $\ell_1$ norm. Makarychev and Shan (2021) and Esfandiari, Mirrokni, and Narayanan (2022) improved the upper bound to $\tilde{O}(\log k)$; see also Laber and Murtinho (2021) and Gamlath, Jia, Polak, and Svensson (2021) for related results. The approximation factor was later improved to $O(\log k)$ by Gupta, Pittu, Svensson, and Yuan (2023) and Makarychev and Shan (2023). Finally, Gupta et al. (2023) established a tight upper bound of $(1 + H_{k-1})$ for the $\ell_1$ norm, where $H_{k-1}$ denotes the $(k-1)$st harmonic number. Bandyapadhyay, Fomin, Golovach, Lochet, Purohit, and Simonov (2022) developed fixed-parameter tractable algorithms that compute the optimal explainable $k$-medians clustering under the $\ell_1$ norm in time $(nd)^{k+O(1)}$ and $n^{2d}(nd)^{O(1)}$. They also proved that the problem is NP-complete and cannot be solved in $f(k)n^{o(k)}$ time for any computable function $f(\cdot)$ unless the Exponential Time Hypothesis (ETH) fails. Gupta et al. (2023) showed that this problem is hard to approximate better than $(1/2 - o(1)) \ln k$ unless P=NP.

Beyond the $\ell_1$ case, much less was known. For $p > 1$, the only prior result was due to Makarychev and Shan (2021), who provided a $\tilde{O}(\log^{3/2} k)$-competitive algorithm and a lower bound of $\Omega(\log k)$ for the $\ell_2$ norm. In this paper, we extend the study of explainable $k$-medians clustering to general $\ell_p$ norms with finite $p \geq 1$. Specifically, we design an algorithm that constructs a threshold decision tree with $k$ leaves, such that the cost of the resulting clustering satisfies

$$\mathbf{E}[\text{cost}_p(X, \mathcal{T})] \leq O(p \cdot \log^{1+1/p-1/p^2} k \cdot \log \log k) \cdot \text{OPT}_{k,p}(X).$$

This improves upon the best known bound for $p = 2$, and for $p = 1$ it matches the optimal guarantee up to an $O(\log \log k)$ factor. Note that the exponent of the logarithm, $1 + 1/p - 1/p^2$, always lies in the interval $[1, 1.25]$.

We now discuss the second contribution of the paper. In recent years, researchers have turned their attention to dynamic clustering algorithms, which maintain a high-quality clustering as the dataset evolves and is continuously updated. Recent work in this area includes papers by Lattanzi and Vassilvitskii (2017); Chan, Guerqin, and Sozio (2018); Cohen-Addad, Hjuler, Parotsidis, Saulpic, and Schwiegelshohn (2019); Deng, Li, and Rabani (2022); Bhattacharya, Costa, Lattanzi, and Parotsidis (2023); Bhattacharya, Costa, Garg, Lattanzi, and Parotsidis (2024); Bhattacharya, Costa, and Farokhnejad (2025).

Dynamic algorithms are typically evaluated based on two key metrics: the update time for insertions and deletions, and the *recourse*—the number of changes made to the solution (in this case, centers inserted or deleted) in response to each update. Bhattacharya et al. (2025) presented an approximation algorithm with $O(1)$-approximation ratio, $O(\log^2 \Delta)$ recourse and $\tilde{O}(k)$ update time (where $\Delta$ is an aspect ratio of the metric space).

In this paper, we initiate the study of dynamic algorithms for *explainable $k$-medians clustering*. Specifically, we ask whether our explainable algorithm can be combined with state-of-the-art dynamic $k$-medians clustering algorithms—and we answer this question affirmatively.

Most known algorithms for explainable $k$-medians clustering first compute a clustering using an existing off-the-shelf method, which we refer to as the *reference clustering*, and then use it to construct a decision tree. Importantly, this second step is *oblivious* to the dataset–that is, it relies only on the reference clustering and not on the actual data points. Our algorithm is no exception: it takes as input a set of reference centers and outputs a threshold decision tree whose cost is upper bounded by $\tilde{O}(p \cdot \log^{1+1/p-1/p^2} k)$ times the cost of the reference clustering. However, existing algorithms for explainable clustering are not designed to operate in a dynamic setting.

We present a dynamic implementation of our algorithm, in which the set of reference centers evolves over time through insertions and deletions. Our algorithm supports updates in $O(d \log^3 k)$ time and modifies only $O(\log k)$ nodes in the tree per update (i.e., it has $O(\log k)$ recourse), while maintaining the same $\tilde{O}(p \cdot \log^{1+1/p-1/p^2} k)$ competitive ratio.

Our algorithm can be integrated with the dynamic algorithms for unconstrained $k$-medians mentioned above. We begin by updating the set of centers using one of these low-recourse algorithms, and then apply our dynamic algorithm to update the decision tree for explainable clustering. Our algorithm can also be used to construct explainable clusterings for multiple values of $k$ – for example, when selecting a suitable $k$ within a given range using the elbow method. In such cases, we can run an algorithm (such as $k$-means++) that outputs centers incrementally, and feed these centers into our dynamic algorithm, which updates the decision tree on the fly.

## 1.1 Techniques

Our static algorithm for explainable $k$-medians under the $\ell_p$ norm builds on and refines a prior algorithm by Makarychev and Shan (2021) developed for the $\ell_2$ norm. In this work, we generalize the approach to all $\ell_p$ norms with finite $p \geq 1$ and provide a tighter analysis. In particular, for the $\ell_2$ norm, we improve the competitive ratio from the previous bound of $\tilde{O}(\log^{1.5} k)$ to $\tilde{O}(\log^{1.25} k)$.

As we noted earlier, our algorithm takes as input a set of reference centers produced by an off-the-shelf clustering algorithm and does not access the dataset points directly.

This algorithm relies on the PARTITION_LEAF procedure. Each call to PARTITION_LEAF takes a cell of the space containing some subset of centers $C_u$ and constructs a partial threshold decision tree that partitions the cell into several subcells, each containing at most a $\tilde{\gamma}$ fraction of the input centers, where $\tilde{\gamma} < 1$. We apply PARTITION_LEAF recursively, starting with the cell containing all centers $c_1, \ldots, c_k$, to construct the full decision tree.

PARTITION_LEAF first selects an *anchor* point within the cell. This anchor, denoted $m^u$, is the median or an approximate median of the centers in $C_u$ and remains fixed throughout the execution of PARTITION_LEAF. The procedure partitions the space using random cuts drawn from a specially crafted distribution. Each time a cut is sampled and applied (some cuts may be discarded), the algorithm removes the centers that are separated from the anchor and places them into one of the output parts. Each cut is defined by a coordinate $i$ and a threshold $\theta$, and has the form $Left = \{x : x_i < \theta\}$ and $Right = \{x : x_i \geq \theta\}$. If a sampled cut does not separate any centers, it is discarded.

Random cuts in the algorithm are drawn as follows: PARTITION_LEAF selects a random coordinate $i \in \{1, \ldots, d\}$, a random threshold $\theta' \in [0, R_t]$, and a random sign $\sigma \in \{\pm 1\}$ (where $R_t$ is the radius of the cell; see Section 2 for details). It lets $\theta = m_i^u + \sigma \theta'$. The cumulative density function for $\theta'$ is given by $x^p / R_t^p$. The algorithm terminates when fewer than $\gamma n$ centers remain unseparated from the anchor.

We note that using a uniform distribution for $\theta$ (i.e., selecting a random coordinate $i$ and then choosing a threshold $\theta$ uniformly at random from $[-R_t, R_t]$) would result in a poor competitive ratio, as illustrated in the following example. Consider a $k$-medians clustering with the $\ell_p$ norm, defined by $k + 1$ centers located at the positions $e_1, \ldots, e_k$, and 0, where $e_i$ denotes the $i$-th standard basis vector. We focus on a single data point $x$ with coordinates $(\varepsilon, \ldots, \varepsilon)$. Suppose we pick cuts by selecting a random coordinate $i \in \{1, \ldots, d\}$ and a threshold $\theta \in [0, 1]$ uniformly at random. In this case, a constant fraction of the centers will be separated from the anchor $m^u$ in $\Theta(k)$ steps. The probability that one of the cuts made during these steps separates $x$ from its closest center (the center located at the origin) is $\Theta(\varepsilon k)$, assuming $\varepsilon$ is sufficiently small. If $x$ is separated from 0, it will be assigned to a different center, i.e., one of the vectors $e_i$. In that case, the $\ell_p$ distance from $x$ to the new center is approximately 1. Therefore, the expected cost of the clustering produced by this variant of the algorithm for point $x$ is $\Theta(\varepsilon k)$, while the optimal (unconstrained) cost is $\varepsilon k^{1/p}$. Hence, the competitive ratio of such an algorithm is at least $\Theta(k^{1-1/p})$.

In this paper, we prove – through a careful analysis of the algorithm – that the aforementioned choice of random distribution yields an $O(p \log^{1+1/p-1/p^2} \log \log k)$ upper bound on the algorithm's competitive ratio.

We then show how to implement our static clustering algorithm in the dynamic setting. Our approach builds on the idea of assigning each decision node a timestamp drawn from an exponential distribution – a technique previously introduced in Gupta et al. (2023); Makarychev and Shan (2023) solely for the purpose of analyzing an explainable clustering algorithm under the $\ell_1$ norm. We extend this idea by integrating the exponential clock directly into the algorithm's design. Specifically, we assume that random cuts are selected with arrival rates governed by a Poisson process. Each cut is assigned a timestamp corresponding to its selection time.

The high-level idea behind the dynamic algorithm is as follows. When a new center is inserted, we identify the earliest cut – based on its timestamp – that separates the new center from the anchor. To efficiently find such a cut, we employ data structures that enable this operation in $O(d \log k)$ time. We prove that this earliest cut corresponds to the one that would have been used by the static algorithm to separate the center $c$ from the anchor $m^u$. There are two possible cases: either the decision tree already contains a node corresponding to this cut, or it does not. In the latter case, the algorithm creates a new decision node to incorporate the cut.

Implementing this idea presents several challenges. The dynamic PARTITION_LEAF algorithm is not permitted to modify the anchor; consequently, it may need to rebuild the entire decision tree for a cell and its descendants once the number of updates in that cell exceeds a certain threshold. Moreover, the dynamic algorithm must terminate at a fixed time–one that cannot be adjusted as centers are added or removed. As a result, unlike the static version, it cannot stop based on the number of remaining centers falling below a given threshold. In this paper, we address these challenges and present a complete dynamic algorithm for the problem.

## 2 Algorithm

In this section, we present our algorithm for constructing an explainable clustering tree for the $k$-medians problem in $\ell_p$ space. The algorithm takes a set of $k$ centers $C$ as input and produces a binary threshold tree $\mathcal{T}$ with $k$ leaves, each leaf containing a distinct center in $C$. The construction begins by initializing the root node $r$ of the tree with all centers $C$, and recursively partitioning the centers using the procedure PARTITION_LEAF (as shown in Figure 1). We initiate the construction by calling PARTITION_LEAF$(r)$.

While this algorithm is static, we show an efficient dynamic algorithm that achieves the same behavior as this algorithm in Section 5. To couple the dynamic algorithm with the static algorithm, we present our algorithm based on two oracles: STOPPING_ORACLE and GET_ANCHOR. The STOPPING_ORACLE takes a cut $\omega$ and the current subtree $\mathcal{T}_u$ rooted at $u$ as input and outputs a Boolean value; if it is True, then it stops partitioning centers; otherwise, the algorithm continues to partition centers. This oracle guarantees that when partitioning stops, every leaf in $\mathcal{T}_u$ contains at most a $\tilde{\gamma}$ fraction of centers in $C_u$, where $\tilde{\gamma} < 1$. The oracle GET_ANCHOR takes a subset of centers $C_u$ and returns an anchor point $m^u \in \mathbb{R}^d$ such that for each coordinate $i \in [d]$, at least $1/4$ of centers in $C_u$ lie on either side of $m_i^u$, i.e. $|\{c \in C_u : c_i \geq m_i^u\}| \geq |C_u|/4$ and $|\{c \in C_u : c_i < m_i^u\}| \geq |C_u|/4$. In the static version, we can simply choose the anchor $m^u$ as the

coordinate-wise median of $C_u$, and the STOPPING_ORACLE returns True if and only if the main part contains fewer than $1/2$ of centers in $C_u$, i.e. $|C_{u_0}| < |C_u|/2$.

---

**Algorithm** PARTITION_LEAF

**Input:** a node $u$ with a set of centers $C_u \subseteq \mathbb{R}^d$
**Output:** a threshold tree $\mathcal{T}_u$

1. Set the anchor $m^u = \text{GET\_ANCHOR}(C_u)$.

2. Set the main part $u_0 = u$ and $C_{u_0} = C_u$ and step $t = 0$. Set the subtree $\mathcal{T}_u$ to have only the root $u$.

3. We iteratively sample cuts $\omega_t$ until STOPPING_ORACLE$(\omega_t, \mathcal{T}_u)$ returns True:

   (a) Update $t = t + 1$ and the radius $R_t = \max_{c \in C_{u_0}} \|c - m^u\|_p$.

   (b) Sample a threshold cut $\omega_t = (i_t, \vartheta_t)$ as follows. Sample $i_t \in \{1, 2, \cdots, d\}$, $\sigma_t \in \{-1, 1\}$, and $(\theta_t)^p \in [0, (R_t)^p]$ uniformly at random. Then, set $\vartheta_t = m^u_{i_t} + \sigma_t \theta_t$.

   (c) **If** the cut $\omega_t$ separates any two centers in $C_{u_0}$, **then**
      - Add two new children $u_L, u_R$ to the main part $u_0$ and split the centers into two parts $C_{u_L} = \{c \in C_{u_0} : c_{i_t} < \vartheta_t\}$ and $C_{u_R} = \{c \in C_{u_0} : c_{i_t} \geq \vartheta_t\}$.
      - Update the main part $u_0 = u_L$ and $C_{u_0} = C_{u_L}$ if $\sigma_t = 1$; otherwise $u_0 = u_R$ and $C_{u_0} = C_{u_R}$ ($u_0$ always contains $m^u$).

4. Call PARTITION_LEAF$(v)$ for each leaf $v$ containing more than one center in the subtree rooted at $u$.

5. Return the tree $\mathcal{T}_u$ rooted at node $u$.

---

Figure 1: Algorithm PARTITION_LEAF for explainable $k$-medians in $\ell_p$

We now describe the procedure PARTITION_LEAF$(u)$. The procedure PARTITION_LEAF$(u)$ operates on a node $u$ that contains a set of centers $C_u$. It first queries the oracle GET_ANCHOR to get an anchor point $m^u$. We always refer to the leaf that contains $m^u$ as the *main part*, and denote it by $u_0$. Initially, we set $u_0 = u$.

PARTITION_LEAF iteratively splits the subset $C_u$ using randomized threshold cuts until the STOPPING_ORACLE returns True. In each iteration $t$, it computes the maximum $\ell_p$ distance from $m^u$ to any center in the current main part $C_{u_0}$, denoted by $R_t = \max_{c \in C_{u_0}} \|c - m^u\|_p$. Then, it samples a random threshold cut $\omega_t$ as follows. A coordinate $i_t \in \{1, 2, \cdots, d\}$ and a sign $\sigma_t \in \{-1, 1\}$ are chosen uniformly at random. Next, it draws a random variable $Z_t$ uniformly from the interval $[0, (R_t)^p]$ and set $\theta_t = (Z_t)^{1/p}$. The resulting threshold cut is $\omega_t = (i_t, \vartheta_t)$, where $\vartheta_t = m^u_i + \sigma_t \cdot \theta_t$. If this threshold cut separates at least two centers in $C_{u_0}$, the algorithm partitions the current main part $u_0$ into two disjoint cells. It adds two children $u_L, u_R$ to the node $u_0$ and assigns centers $C_{u_L} = \{c \in C_{u_0} : c_{i_t} < \vartheta_t\}$ to node $u_L$ and centers $C_{u_R} = \{c \in C_{u_0} : c_{i_t} \geq \vartheta_t\}$ to node $u_R$. The child node, either $u_L$ or $u_R$, that contains anchor $m^u$ becomes the updated main part $u_0$. This process continues until the STOPPING_ORACLE returns True. Finally, it recursively calls the PARTITION_LEAF$(v)$ on each leaf $v$ that contains more than one center in the subtree rooted at $u$.

## 3 Analysis of approximation factor

In this section, we provide the approximation guarantees for our algorithm.

**Theorem 3.1.** *Given a set of points $X$ and a set of $k$ centers $C$, for any $p \geq 1$, Algorithm finds a threshold tree $\mathcal{T}$ with $k$ leaves that has $k$-medians cost*

$$\mathbf{E}[\text{cost}_p(X, \mathcal{T})] \leq O\left(p \cdot (\log k)^{1 + \frac{1}{p} - \frac{1}{p^2}} \log \log k\right) \text{cost}_p(X, C).$$

We analyze the approximation guarantee by bounding the expected cost incurred by each point $x \in X$. Fix an arbitrary point $x \in X$ and let $c \in C$ be its closest center. We show that the expected cost of

assigning $x$ in the constructed threshold tree $\mathcal{T}$ is bounded by

$$\mathbf{E}[\text{cost}_p(x, \mathcal{T})] \leq O\left(p \cdot (\log k)^{1+\frac{1}{p}-\frac{1}{p^2}} \log \log k\right) \|x - c\|_p. \tag{1}$$

If $x$ equals its closest center $c$, then $x$ is always assigned to $c$ by any tree $\mathcal{T}$, and thus incurs zero cost, $\text{cost}_p(x, \mathcal{T}) = 0$. In this case, the above bound holds trivially. Therefore, we may assume from now on that $x \neq c$.

Consider the path from the root to the leaf in the tree that contains this point $x$. We index the node on this path by $t = 1, 2, \cdots, T$, where $u_1$ is the root of the tree and $u_T$ is the leaf that contains $x$. Let $\mathcal{T}_t$ be the partially built tree when the node $u_t$ is generated in the algorithm. Given any tree $\mathcal{T}_t$, let $\mathcal{T}_t(x)$ be the closest center in the same leaf as $x$ in tree $\mathcal{T}_t$. We define the following upper bound on the approximation factor.

**Definition 3.2.** *Let $A_k$ be the smallest number such that the following inequality holds for every partially built tree $\mathcal{T}_t$,*
$$\mathbf{E}\left[\text{cost}_p(x, \mathcal{T}) \mid \mathcal{T}_t\right] \leq A_k \cdot \|x - \mathcal{T}_t(x)\|_p.$$

Since all centers are contained in the root $u_1$, we have $\mathcal{T}_1(x) = c$. Thus, we have $A_k$ is an upper bound on the approximation factor. We then prove the following lemma, which provides a recurrence relation for bounding $A_k$.

**Lemma 3.3.** *For some absolute constant $\beta > 0$, we have for any step $t^*$*

$$\mathbf{E}\left[\frac{\text{cost}_p(x, \mathcal{T})}{\|x - \mathcal{T}_{t^*}(x)\|_p} \mid \mathcal{T}_{t^*}\right] \leq 3 + \frac{2A_k}{k} + \beta \cdot p(\log k)^{1+\frac{1}{p}-\frac{1}{p^2}} \cdot \log(A_k \log^2 k).$$

We first show how to use Lemma 3.3 to get the desired bound on $A_k$, which also provides the approximation factor for the algorithm.

*Proof of Theorem 3.1.* By Lemma 3.3 and the definition of $A_k$, we get the following recurrence relation on $A_k$, $A_k \leq 3 + \frac{2A_k}{k} + \beta \cdot p(\log k)^{1+\frac{1}{p}-\frac{1}{p^2}} \cdot \log(A_k \log^2 k)$. Then, we have that $A_k$ is bounded by $A_k \leq O\left(p(\log k)^{1+\frac{1}{p}-\frac{1}{p^2}} \log \log k\right)$. By the definition of $A_k$, we bound the expected cost of any point $x \in X$ given by tree $\mathcal{T}$ as shown in Equation (1). By taking the sum over all points in $X$, we get the approximation factor for the algorithm. $\qquad\square$

## 3.1 Radius and diameter bounds

Before proving the main recurrence lemma, we establish several key results that describe how the radius and diameter of clusters evolve during the recursive partitioning process. These results serve as essential tools in our main proof. We defer the proofs to Appendix A.1.

We first show that the radius $R_u$ decreases exponentially in one partition leaf call. Consider any partition leaf call on a node $u$. Let $R_t$ be the radius of the main part before the iteration $t$ of this partition leaf call. Then, we have $R_1 = R_u$. We use $\mathcal{T}_t$ to denote the partial tree given by the algorithm before the iteration $t$ of this partition leaf call.

**Lemma 3.4.** *Consider any partition leaf call on node $u$. Let $L = \lceil 2^{p+3} d \ln k \rceil$. Then for every $t \geq 1$, we have $\Pr\{R_{t+L} > R_t/2 \mid \mathcal{T}_t\} \leq \frac{1}{k^3}$.*

We define the diameter of a node $u$ to be $D_u := \max_{c,c' \in C_u} \|c - c'\|_p$. We use the following relation between $R_u$ and $D_u$ for a node $u$ at the beginning of a partition leaf call, which generalizes Lemma 6.1 in Makarychev and Shan (2022) to $\ell_p$ norm.

**Lemma 3.5** (Lemma 6.1 in Makarychev and Shan (2022)). *For every node $u$ on which the algorithm calls partition leaf, we have $R_u/4^{1/p} \leq D_u \leq 2R_u$.*

We define $\widetilde{D}_u$ for every node $u$ as follows. If the algorithm calls partition leaf on node $u$, then $\widetilde{D}_u = D_u$. Now consider any node $v$ in the partition leaf call of a node $u$, on which the algorithm does not call the partition leaf. Let $d(u, v)$ be the distance from $v$ to $u$ in the tree. We set $\widetilde{D}_v = \max\left\{D_v, \widetilde{D}_u \cdot \frac{R_v}{R_u}\right\}$. By the definition, $\widetilde{D}_u$ is an upper bound of the diameter $D_u$ for every node

$u$. We now show that $\widetilde{D}_u$ is non-increasing along any path from the root to a leaf in the tree. Since $R_v$ is non-increasing in one partition leaf call, $\widetilde{D}_v$ is also non-increasing in one partition leaf call. Moreover, since $\widetilde{D}_v \geq D_v$ for every node $v$ and $\widetilde{D}_u = D_u$ on node $u$ where the algorithm calls partition leaf, we have $\widetilde{D}_v$ is also non-increasing across partition leaf calls.

**Lemma 3.6.** *For every node $u$, we have $R_u/4^{1/p} \leq \widetilde{D}_u \leq 2R_u$.*

We then show that $\widetilde{D}_u$ decreases exponentially along any path from the root to a leaf in the tree.

**Lemma 3.7.** *Let $L' = \lceil 2^{2p+6} d \ln k \rceil$. For every node $u$, let node $v$ be any descendant of $u$ at depth $L'$ in the tree $\mathcal{T}$. Then, we have $\Pr\{\widetilde{D}_v \geq \widetilde{D}_u/2 \mid \mathcal{T}_u\} \leq \frac{4}{k^3}$.*

## 3.2 Recurrence lemma

In this section, we provide a proof overview of Lemma 3.3, which establishes the recurrence relation of $A_k$. The details of the proof are deferred to Appendix A.2.

We fix an arbitrary point $x \in X$. Without loss of generality, we consider the step $t^* = 1$ and then $\mathcal{T}_{t^*}(x) = c$ is the closest center to $x$ in $C$. We then focus on the nodes in $\mathcal{T}$ that contain this point $x$, which form a path from the root to the leaf containing $x$. We index the node along this path by step $t = 1, 2, \cdots, T$, where $u_1$ is the root of the tree and $u_T$ is the leaf that contains $x$. Let $\mathcal{T}_t$ be the partially built tree when the node $u_t$ is generated in the algorithm.

We now bound the cost of this point $x$ given by the tree $\mathcal{T}$. We begin by assuming that the radius $R_t$ and the diameter substitute $\widetilde{D}_t$ decrease by a factor of 2 after every $L$ and $L'$ steps, respectively. By Lemma 3.4 and 3.7, and applying the union bound over all iterations, this good event holds with probability at least $1 - 1/k$. If this good event fails to hold, then we simply upper bound the expected cost of $x$ by $A_k\|x - c\|_p$, which contributes the $A_k/k$ factor.

Consider a node $u_t$ such that both $x$ and $c$ are contained in $u_t$, and let $\omega_t$ be the cut sampled at this node. Let $C_t$ be the set of centers contained in $u_t$ and $D_t$ be the diameter of $u_t$. If $x$ and $c$ are separated by this cut $\omega_t$, then $x$ is eventually assigned to a different center in $C_t$ by $\mathcal{T}$. By the triangle inequality, we have the cost of $x$ in $\mathcal{T}$ is at most $\|x - c\|_p + D_t$. Alternatively, we can use a more refined bound based on the notion of the fallback center, following the approach in Makarychev and Shan (2021, 2022). If $x$ is separated from $c$ by this cut $\omega_t$, then we define the fallback center of $x$ to be the closest center $c' \in C_{t+1}$ to $x$ that is not separated from $x$ by this cut $\omega_t$. This fallback center depends on the tree $\mathcal{T}'$ and the cut $\omega_t$. Let $M_t(\omega_t)$ denote the distance fro m $x$ to the fallback center. Then, by the definition of $A_k$, the expected cost of $x$ can also be upper bounded by $A_k M_t(\omega_t)$.

We now partition the steps $\{1, 2, \cdots, T\}$ into three disjoint cases based on the radius $R_t$ and the fallback distance $M_t(\omega)$ as follows. We introduce the following definitions.

**Definition 3.8.** *For a fixed parameter $\alpha > 0$, we say that step $t$ is a light step if the radius satisfies*

$$R_t \leq 6 \log^\alpha k \cdot \max\left\{\|x - m^t\|_p, \|c - m^t\|_p\right\}.$$

*Otherwise, step $t$ is called a heavy step.*

If $x$ and $c$ are separated by a cut $\omega_t$, then we refer to this cut as a light cut if step $t$ is a light step, and a heavy cut if step $t$ is a heavy step.

**Definition 3.9.** *For each step $t$, we say a cut $\omega_t$ separating $x$ and $c$ a safe cut if $A_k M_t(\omega_t) \leq \frac{R_t}{6^p \log^2 k}$. Otherwise, this cut $\omega_t$ is called an unsafe cut.*

Therefore, if $x$ and $c$ are separated by the tree $\mathcal{T}$, then exactly one of the following three events must occur: (1) they are separated by a safe cut; (2) they are separated by a light cut; (3) they are separated by a heavy and unsafe cut. We then show how to bound the contribution of each case to the expected cost separately.

**Safe cut:** Suppose $x$ and $c$ are contained in node $u_t$. The probability that $x$ and $c$ are separated by the cut $\omega_t$ is at most

$$\Pr\{x \ \& \ c \text{ separated by } \omega_t \mid \mathcal{T}_t\} \leq \frac{1}{2d} \cdot \frac{p\|x - c\|_p(\|x - m^t\|_p^{p-1} + \|c - m^t\|_p^{p-1})}{R_t^p}.$$

In this case, we use $A_k M_t$ as the upper bound of the expected cost since it is much smaller than the radius $R_t$. We show that $3R_t \geq \|x - m^t\|_p + \|c - m^t\|_p$. Thus, the expected cost of a safe cut at step $t$ is at most $\frac{p}{2d} \cdot \frac{A_k M_t}{R_t} \cdot 3^{p-1} \cdot \|x - c\|_p$. In each partition leaf call, we know that $M_t$ is non-decreasing as $t$ increases and $R_t$ decreases by a factor of 2 after every $L$ steps. Hence, $A_k M_t / R_t$ forms an increasing geometric series in every $L$ steps. Since $A_k M_t / R_t \leq 1/(6^p \log^2 k)$ for safe cuts, the expected cost due to safe cuts in one partition leaf call is at most

$$L \cdot \frac{p}{2d} \cdot \frac{2}{6^p \log^2 k} \cdot 3^{p-1} \cdot \|x - c\|_p \leq O\left(\frac{1}{\log k}\right) \|x - c\|_p.$$

Combining over all $O(\log k)$ partition leaf calls, this case is bounded by $O(1) \cdot \|x - c\|_p$.

**Light cut:** Consider the node $u_t$ contains $x$ and $c$. The probability that $x$ or $c$ is separated from the anchor $m^t$ by $\omega_t$ is at least

$$\Pr\{x \text{ or } c \text{ separated from } m^t \text{ by } \omega_t \mid \mathcal{T}_t\} \geq \frac{1}{2d} \cdot \frac{\max\{\|x - m^t\|_p^p, \|c - m^t\|_p^p\}}{R_t^p}.$$

Thus, in each partition leaf call, the probability that $x$ and $c$ are separated by a light cut at the end of the partition leaf call is most

$$\frac{p\|x - c\|_p(\|x - m^t\|_p^{p-1} + \|c - m^t\|_p^{p-1})}{\max\{\|x - m^t\|_p^p, \|c - m^t\|_p^p\}}.$$

We upper bound the expected penalty by $D_t \leq 2R_t \leq 12 \log^\alpha k \cdot \max\{\|x - m^t\|_p, \|c - m^t\|_p\}$ by the definition of a light cut. Since the number of partition leaf calls is at most $O(\log k)$, the expected cost due to a light cut is at most

$$O(\log k) \cdot D_t \cdot \frac{p\|x - c\|_p(\|x - m^t\|_p^{p-1} + \|c - m^t\|_p^{p-1})}{\max\{\|x - m^t\|_p^p, \|c - m^t\|_p^p\}} \leq O(p \log^{1+\alpha} k)\|x - c\|_p.$$

**Heavy and unsafe cut:** Consider a heavy step $t$ when $x$ and $c$ are contained in node $u_t$. For each coordinate $i$, we define $U_i(t) = \{\vartheta : (i, \vartheta) \text{ is unsafe}\}$ to be all thresholds $\vartheta$ such that the cut $\omega_t = (i, \vartheta)$ is unsafe at step $t$. Let $\delta_i(t)$ be the Lebesgue measure of the unsafe threshold $U_i(t)$. Then, the probability that $x$ and $c$ are separated by an unsafe cut at the heavy step $t$ is at most

$$\frac{p}{2d} \cdot \sum_{i=1}^{d} \frac{\max\{|x_i - m_i^t|, |c_i - m_i^t|\}^{p-1}}{R_t^p} \cdot \delta_i(t).$$

Note that all steps in $P_s$ in one partition leaf call $s$ uses the same anchor point $m^s$. Let $P_s' \subseteq P_s$ be all heavy steps in the partition leaf call. We define a vector $\Delta(s) \in \mathbb{R}^d$ whose $i$-th coordinate is $\Delta_i(s) = \sum_{t \in P_s'} \delta_i(t)$. By summing the above separation probability over all steps in $P_s'$ and applying Hölder's inequality, the probability that $x$ and $c$ are separated by a heavy and unsafe cut in partition leaf call $s$ is at most

$$\frac{p}{2d} \cdot \|\Delta(s)\|_p \cdot \frac{\|x - m^s\|_p^{p-1} + \|c - m^s\|_p^{p-1}}{R_t^p}.$$

In this case, we upper bound the penalty of separation by $D_t \leq 2R_t$. Since $R_t \geq 6 \log^\alpha k \cdot \max\{\|x - m^t\|_p, \|c - m^t\|_p\}$ for heavy steps, we have the expected penalty due to heavy and unsafe cuts is at most

$$\frac{p}{d} \cdot \frac{2}{(6 \log^\alpha k)^{p-1}} \cdot \sum_{s=1}^{S} \|\Delta(s)\|_p.$$

We then bound $\sum_{s=1}^{S} \|\Delta(s)\|_p$. Since the number of partition leaf calls is $S = O(\log k)$, we show that

$$\sum_{s=1}^{S} \|\Delta(s)\|_p \leq \log^{1-\frac{1}{p}} k \left\|\sum_{s=1}^{S} \Delta(s)\right\|_p.$$

Consider any fixed cut $\omega = (i, \vartheta)$ that separates $x$ and $c$. This cut is unsafe at step $t$ if and only if $M_t(\omega) \geq R_t/(6^p \log^2 k \cdot A_k)$. Moreover, it always holds $M_t(\omega) \leq D_t$. By Lemma 3.6, we

have $R_t \geq \widetilde{D}_t$ and $\widetilde{D}_t \geq D_t$. Since by Lemma 3.7, $\widetilde{D}_t$ decreases by a factor of 2 after every $L' = \lceil 2^{2p+6} d \ln k \rceil$ steps, this cut $\omega$ is unsafe in at most $L' \cdot \log(2 \cdot 6^p \log^2 k \cdot A_k)$ steps. Thus, we have

$$\left\| \sum_{s=1}^{S} \Delta(s) \right\|_p \leq O(4^p \cdot d \log k \cdot p \log(\log^2 k \cdot A_k)) \|x - c\|_p.$$

Therefore, the expected cost due to heavy and unsafe cuts is at most

$$O\left( (\log k)^{2 - \frac{1}{p} - \alpha(p-1)} \log(\log^2 k \cdot A_k) \right) \|x - c\|_p.$$

Finally, combining all three cases and taking $\alpha = 1/p - 1/p^2$, we get the conclusion.

## 4 Lower bounds

In this section, we present two lower bound results for explainable $k$-medians under $\ell_p$ norms. First, we provide an $\Omega(\log k)$ lower bound on the competitive ratio of explainable $k$-medians under $\ell_p$ norm, for any fixed $p \geq 1$. Second, we show that no explainable clustering algorithm can, without knowing $p$ in advance, achieve a good competitive ratio simultaneously for all $p \geq 1$. In particular, there exists an instance on which any such algorithm incurs a competitive ratio of $\Omega(d^{1/4})$ for some $p \geq 1$.

We extend the lower bound instance for explainable $k$-medians in $\ell_2$ by Makarychev and Shan (2021) to all $\ell_p$ norms with $p \geq 1$. The proof is provided in Appendix D.

**Theorem 4.1.** *For every $p \geq 1$, there exists an instance $X \subseteq \mathbb{R}^d$, such that for every threshold tree $\mathcal{T}$, its clustering cost is at least $\mathrm{cost}_p(X, \mathcal{T}) = \Omega(\log k)\mathrm{OPT}_{k,p}(X)$, where $\mathrm{OPT}_{k,p}(X)$ is the $\ell_p$ cost of the optimal (unconstrained) $k$-medians clustering of $X$.*

The competitive ratio of our algorithm is upper bounded by $\tilde{O}(p(\log k)^{1+1/p-1/p^2})$. Thus, for every $p > 1$, there remains an $\tilde{O}((\log k)^{1/p-1/p^2})$ gap, which is maximized at $p = 2$ as $\tilde{O}(\log^{1/4} k)$.

We then investigate whether it is possible to design an explainable clustering algorithm that, without knowing $p$ in advance, produces a single threshold tree (or a distribution over threshold trees) with a good competitive ratio for all $p \geq 1$ simultaneously. The following theorem shows that this is not possible. The proof is in the Appendix C.

**Theorem 4.2.** *There exists an instance $X \subseteq \mathbb{R}^d$, such that for any distribution over threshold trees, the expected competitive ratio is at least $\Omega(d^{1/4})$ for some $p \geq 1$.*

## 5 Dynamic algorithm

In this section, we present a dynamic algorithm for the setting where the input set of points $X$ and centers $C$ change over time. We show that after each update, our algorithm maintains a threshold tree with low $k$-medians cost and analyze its update time and recourse.

Let $X_1, X_2, \ldots, X_t, \ldots$ denote a changing data set after each update $t$ and let $C_1, C_2, \ldots, C_t, \ldots$ be the corresponding sequence of center sets. Our goal is to output after each update $t$ a threshold tree $\mathcal{T}_t$ with $|C_t|$ leaves that approximates the clustering of $X_t$ with centers $C_t$. Similarly to the static setting, our dynamic algorithm only depends on the center sets to construct the trees $\mathcal{T}_t$. Thus, we focus on the setting where the center sets change through a sequence of *insertion* or *deletion requests*, i.e. $C_t = C_{t-1} \cup \{c\}$, if $t$ is an insertion request of a new center $c$, or $C_t = C_{t-1} \setminus \{c\}$, if $t$ is a deletion request of an existing center $c \in C_{t-1}$. We show the following theorem, with the proof in Appendix B.

**Theorem 5.1.** *Given a sequence of requests, where each request is either an insertion or a deletion of a single center in $\mathbb{R}^d$, there is a dynamic algorithm that for each center set $C_t$, outputs a threshold tree $\mathcal{T}_t$ such that for any data set $X \subseteq \mathbb{R}^d$,*

$$\mathbf{E}[\mathrm{cost}_p(X, \mathcal{T}_t)] \leq O(p \cdot (\log k_t)^{1+1/p-1/p^2} \log \log k_t) \mathrm{cost}_p(X, C_t),$$

*where $k_t = |C_t|$. The amortized update time of the algorithm is $O(d \log^3 k)$ and the amortized recourse (number of tree nodes updated) is $O(\log k)$, where $k = \max_{i=1}^{t} |C_i|$.*

Note that naively classifying a data point $x$ using a threshold tree $\mathcal{T}_t$ takes $O(k)$ time in the worst case, if $\mathcal{T}_t$ has height $O(k)$. In contrast, our dynamic algorithm efficiently updates the current threshold tree in only $O(d \log^3 k)$ time, by modifying on average $O(\log k)$ nodes after each request.

Moreover, our dynamic algorithm extends naturally to the *fully-dynamic* explainable clustering setting, where the input is a stream of insertion or deletion requests of *data points* instead of centers. Specifically, we invoke a fully-dynamic clustering algorithm by Bhattacharya et al. (2025) to maintain a sequence of center sets $C_t$ that provide a constant-factor approximation on $X_t$. Since the algorithm of Bhattacharya et al. (2025) guarantees that only $\tilde{O}(1)$ centers change on average after each update, our dynamic algorithm applies directly by treating each center change as a center update request and invoking Theorem 5.1. See Corollary B.6 for the formal statement.

To implement our dynamic algorithm, we reinterpret the PARTITION_LEAF procedure (Figure 1) in an equivalent but more convenient way using the exponential clock. This version generates all random cuts in advance. Without loss of generality, we assume that all centers lie within $[-1,1]^d$; otherwise, we rescale the instance accordingly. The procedure generates an infinite sequence of candidate cuts $\omega_1, \omega_2, \ldots$, where each cut $\omega_t = (i_t, \vartheta_t)$ is constructed as follows: a coordinate $i_t$, a sign $\sigma_t \in \{-1, 1\}$, and a parameter $Z_t \in [0, 2^p]$ are sampled uniformly at random. The threshold is then set to $\vartheta_t = m_{i_t} + \sigma_t \cdot (Z_t)^{1/p}$, where $m$ denotes the anchor point. Additionally, each cut $\omega_t$ is assigned an *arrival time* $\rho_t$, such that $\rho_1 \leq \rho_2 \leq \ldots$ follows the arrival times of a Poisson Process with rate $\lambda = 1$.

The algorithm attempts the next cut $(\omega_t, \rho_t)$ in the sequence until the STOPPING_ORACLE returns True. If $\omega_t$ separates at least two centers from the main part, the cut is made; otherwise, it is ignored. Since the arrival times $\rho_t$ are independent of cut choices $\omega_t$, this version yields the same distribution of threshold trees as the original PARTITION_LEAF procedure. These arrival times $\rho_t$ are crucial for the design of our dynamic algorithm. In the following discussion, we assume there is a data structure that stores this sequence of cuts with their arrival times. It also provides a function GET_EARLIEST_CUT that takes a center $c$ and returns the earliest cut $\omega$ from the sequence that separates $c$ and the anchor $m$.

We provide a dynamic implementation of the PARTITION_LEAF procedure, which we apply recursively to obtain a fully dynamic version of the entire clustering algorithm. The dynamic variant of PARTITION_LEAF supports three operations: (1) REBUILD, (2) INSERT CENTER, and (3) DELETE CENTER. We now briefly describe each of these operations.

**Rebuild:** Reconstruct the subtree rooted at node $u$, partitioning all centers in $C_u$ into distinct leaves via recursive calls to the PARTITION_LEAF procedure. In particular, GET_ANCHOR$(C_u)$ returns the true coordinate-wise median of the centers $C_u$ and STOPPING_ORACLE$(\omega_t, \mathcal{T}_t)$ returns True if and only if the main part after $\omega_t$ contains at most $|C_u|/2$ centers. The REBUILD operation is initially called for $C_1$. Next, for every node $u$ where a REBUILD has been called, we keep the number of centers $k_u = |C_u|$ contained in $u$ at the timestep it was *last rebuilt*, and also track the number of updates (insertions / deletions) of $u$ since that timestep. If this counter exceeds $k_u/4$, the operation REBUILD is called again at node $u$.

**Insert:** Suppose a new center $c$ is inserted. The algorithm calls GET_EARLIEST_CUT to find the earliest cut $\omega$ in the pre-generated sequence with its arrival time $\rho$ that separates $c$ from the anchor $m^u$. Let $(\omega'_1, \rho'_1), \cdots, (\omega'_r, \rho'_r)$ be the cuts currently used in this partition leaf call. Let $\rho^u$ be the stopping time assigned to this partition leaf call during its most recent rebuild. We consider three cases as follows: (1) $\rho = \rho'_j$ for some $j \in [r]$; (2) $\rho > \rho^u$; (3) $\rho \leq \rho^u$ and $\rho \neq \rho'_j$ for any $j \in [r]$.

Case (1): Assign this new center $c$ to the node $v$ generated by cut $\omega'_j$ and recursively maintain the partition leaf call rooted at $v$.

Case (2): This new center $c$ remains in the main part $u_0$ until this partition leaf call ends. We then recursively maintain the partition leaf call on the main part $u_0$.

Case (3): It finds the smallest index $j \in [r]$ such that $\rho < \rho'_j$ or sets $j = r + 1$ if no such index exists. Then we insert this new cut $\omega$ at position $j$ and add a new leaf node containing $c$ to the tree.

**Delete**: Now suppose a center $c \in C_u$ is deleted. We locate the leaf node containing $c$ in this partition leaf call. If this leaf contains only one center $c$, we remove both the leaf and the cut that created it. Otherwise, we delete $c$ from the leaf and maintain the next partition call recursively.

## Acknowledgments and Disclosure of Funding

K. Makarychev and I. Papanikolaou were supported by the NSF Awards CCF-1955351 and EECS-2216970. We thank the anonymous reviewers for their insightful comments and constructive suggestions.

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

# A Proofs in Section 3

## A.1 Proofs in Section 3.1

**Lemma 3.4.** *Consider any partition leaf call on node $u$. Let $L = \lceil 2^{p+3} d \ln k \rceil$. Then for every $t \geq 1$, we have $\Pr\{R_{t+L} > R_t/2 \mid \mathcal{T}_t\} \leq \frac{1}{k^3}$.*

*Proof of Lemma 3.4.* Let $C_t$ be the centers contained in the main part before the iteration $t$ of the partition leaf call. Then, we have $C_1 = C_u$ be the set of centers contained in node $u$. Let $m^u$ be the median of centers in $C_u$. Consider any center $c \in C_t$ with $\|c - m^u\|_p > R_t/2$. Suppose the algorithm chooses the coordinate $i$ at iteration $t$. Then, this center $c$ is separating from $m^u$ at iteration $t$ if and only if $\sigma_t = \mathrm{sgn}(c_i - m_i^u)$ and $\theta_t \in (0, |c_i - m_i^u|]$. Thus, we have

$$\Pr\{c, m^u \text{ are separated at } t \mid \mathcal{T}_t, i_t = i\} = \frac{1}{2} \frac{|c_i - m_i^u|^p}{R_t^p}.$$

Combining all coordinates, the probability that $c$ is separated from $m^u$ at iteration $t$ is at least

$$\Pr\{c, m^u \text{ are separated at } t \mid \mathcal{T}_t\} = \sum_{i=1}^d \frac{1}{d} \cdot \Pr\{c, m^u \text{ are separated at } t \mid \mathcal{T}_t, i_t = i\}$$

$$= \sum_{i=1}^d \frac{1}{2d} \cdot \frac{|c_i - m_i^u|^p}{R_t^p} = \frac{1}{2d} \frac{\|c - m^u\|_p^p}{R_t^p} \geq \frac{1}{2d \cdot 2^p} = \frac{1}{2^{p+1} d}.$$

Since in one partition leaf call, the radius $R_t$ is non-increasing as $t$ increases, for any iteration $t' \geq t$, we have $\|c - m^u\|_p > R_{t'}/2$. Hence, conditioned on $\mathcal{T}_t$, if $c$ is not separated from $m^u$ before iteration $t' \geq t$, then $c$ is separated from $m^u$ at iteration $t'$ with probability at least $1/2^{p+1}d$. Therefore, the probability that $c$ is not separated from $m^u$ after $L = \lceil 2^{p+3} d \ln k \rceil$ iterations is at most

$$\left(1 - \frac{1}{2^{p+1} d}\right)^L \leq e^{-\frac{L}{2^{p+1} d}} = \frac{1}{k^4}.$$

Since there are at most $k$ centers with distance to $m^u$ greater than $R_t/2$, by the union bound over all such centers, we have

$$\Pr\{R_{t+L} > R_t/2 \mid \mathcal{T}_t\} \leq \frac{1}{k^3}.$$

$\square$

We show the following relation between the radius $R_u$ and the diameter $D_u$ for each node $u$ on which the algorithm calls the partition leaf.

**Lemma 3.5** (Lemma 6.1 in Makarychev and Shan (2022))**.** *For every node $u$ on which the algorithm calls partition leaf, we have $R_u/4^{1/p} \leq D_u \leq 2R_u$.*

*Proof of Lemma 3.5.* It is easy to get the second bound from the triangle inequality of the $\ell_p$ norm. Let $m^u$ be the median of centers in $C_u$. We have for any two centers $c, c' \in C_1^u$,

$$\|c - c'\|_p \leq \|c - m^u\|_p + \|m^u - c'\|_p \leq 2R_u.$$

We then show the first bound. For any function $f : C_u \to \mathbb{R}$, let $\mathrm{avg}_{c \in C_u} f(c) = \frac{1}{|C_u|} \sum_{c \in C_u} f(c)$ be the average of $f(c)$ over all centers in $C_u$. Let $c' = \arg\max_{c \in C_u} \|c - m^u\|_p$ be the center that is farthest from the median $m^u$ in $\ell_p$ norm. For any pair of centers $c, \hat{c} \in C_u$, the distance between $c$ and $\hat{c}$ is at most the diameter of $u$, $\|c - \hat{c}\|_p \leq D_u$. Thus, we have

$$D_u^p \geq \mathrm{avg}_{c \in C_u} \|c' - c\|_p^p = \mathrm{avg}_{c \in C_u} \sum_{i=1}^d |c_i' - c_i|^p = \sum_{i=1}^d \mathrm{avg}_{c \in C_u} |c_i' - c_i|^p.$$

Since $m^u$ is the output of GET_ANCHOR which always returns an approximate median of the centers in $C_u$, at least $\frac{1}{4}$ of the centers $c \in C_u$ lie on the opposite side of the hyperplane $\{x : x_i = m_i^u\}$ from the center $c'$. Thus, for these centers $c \in C_u$, we have $|c_i' - c_i| \geq |c_i' - m_i^u|$. As a result

$$D_u^p \geq \sum_{i=1}^{d} \text{avg}_{c \in C_u} |c_i' - c_i|^p \geq \sum_{i=1}^{d} \frac{1}{4} \cdot |c_i' - m_i^u|^p = \frac{1}{4} \cdot \|c' - m^u\|_p^p = \frac{1}{4} R_u^p,$$

which implies $R_u/4^{1/p} \leq D_u$. $\qquad\square$

**Lemma 3.6.** *For every node $u$, we have $R_u/4^{1/p} \leq \widetilde{D}_u \leq 2R_u$.*

*Proof of Lemma 3.6.* For any node $u$ on which the algorithm calls the partition leaf, we have $\widetilde{D}_u = D_u$. By Lemma 3.5, we have $R_u/4^{1/p} \leq \widetilde{D}_u \leq 2R_u$.

We then consider any node $v$ which is not a partition leaf call node. Let $u$ be the node of partition leaf call that generates the node $v$. Since $\widetilde{D}_u \leq 2R_u$, we have $\widetilde{D}_u R_v/R_u \leq 2R_v$. Note that $D_v \leq 2R_v$. Thus, we have $\widetilde{D}_v \leq 2R_v$. Since $\widetilde{D}_u \geq R_u/4^{1/p}$, we have $\widetilde{D}_v \geq \widetilde{D}_u \cdot R_v/R_u \geq R_v/4^{1/p}$. $\qquad\square$

We then show that $\widetilde{D}_u$ decreases exponentially along any path from the root to a leaf in the tree. First, we show that any pair of centers that are far apart in the node are separated with high probability. Let $\mathcal{T}_u$ be the partial tree when node $u$ is generated in the algorithm.

**Lemma A.1.** *For every two centers $c'$ and $c''$ in $C_u$ at distance at least $\widetilde{D}_u/2$,*

$$\Pr\{c', c'' \text{ are separated at } u \mid \mathcal{T}_u\} \geq \frac{1}{d \cdot 2^{2p+2}}.$$

*Proof.* Suppose the algorithm picks coordinate $i$ at node $u$. For every two centers $c', c'' \in C_u$, we consider the following two cases: (1) $c'$ and $c''$ are on the same side of the median $m^u$ in coordinate $i$; (2) $c'$ and $c''$ are on the opposite side of the median $m^u$ on coordinate $i$.

For the first case, without loss of generality, we assume that $c_i'' \geq c_i' \geq m_i^u$. Then, two centers $c'$ and $c''$ are separated by the cut at node $u$ if and only if the algorithm picks $\sigma_u = 1$ and $\theta_u \in (c_i' - m_i^u, c_i'' - m_i^u]$. Let $\mathcal{T}_u$ be the partial tree when node $u$ is generated. Then, we have

$$\Pr\{c', c'' \text{ are separated at } u \mid i_u = i, \mathcal{T}_u\} = \frac{1}{2} \cdot \frac{(c_i'' - m_i^u)^p - (c_i' - m_i^u)^p}{R_u^p}$$
$$\geq \frac{(c_i'' - c_i')^p}{2R_u^p},$$

where the inequality is because $x^p$ is convex and increasing on $[0, \infty)$.

For the second case, $c_i'$ and $c_i''$ are on the opposite side of $m_i^u$. Assume that $c_i' \geq m_i^u \geq c_i''$. Thus, centers $c'$ and $c''$ are separated by the cut at node $u$ if and only if $\sigma = +1, \theta \in (0, c_i' - m_i^u]$ or $\sigma = -1, \theta \in (0, c_i'' - m_i^u]$. Thus, we have

$$\Pr\{c', c'' \text{ are separated at } u \mid i_u = i, \mathcal{T}_u\} = \frac{1}{2} \cdot \frac{|c_i'' - m_i^u|^p + |c_i' - m_i^u|^p}{R_u^p}$$
$$\geq \frac{|c_i'' - c_i'|^p/2^{p-1}}{2R_u^p},$$

where the inequality is from $(a^p + b^p)/2 \geq ((a + b)/2)^p$ for $a, b \geq 0$ since $x^p$ is convex on $[0, \infty)$.

Combining all coordinates, we have the probability that $c'$ and $c''$ are separated at node $u$ is at least

$$\Pr\{c', c'' \text{ are separated at } u \mid \mathcal{T}_u\} \geq \sum_{i=1}^{d} \frac{1}{d} \cdot \frac{|c_i'' - c_i'|^p}{(2R_u)^p} \geq \frac{\|c'' - c'\|_p^p}{d(2R_u)^p}.$$

Since $\widetilde{D}_u \geq R_u/4^{1/p}$, we have for every two centers $c', c'' \in C_u$ with $\|c'' - c'\|_p \geq \widetilde{D}_u/2$,

$$\Pr\{c', c'' \text{ are separated at } u \mid \mathcal{T}_u\} \geq \frac{R_u^p}{4 \cdot 2^p} \cdot \frac{1}{d(2R_u)^p} = \frac{1}{2^{2p+2}d}.$$

$\qquad\square$

**Lemma 3.7.** *Let $L' = \lceil 2^{2p+6} d \ln k \rceil$. For every node $u$, let node $v$ be any descendant of $u$ at depth $L'$ in the tree $\mathcal{T}$. Then, we have $\Pr\{\widetilde{D}_v \geq \widetilde{D}_u/2 \mid \mathcal{T}_u\} \leq \frac{4}{k^3}$.*

*Proof of Lemma 3.7.* Let $u'$ be the node at which the algorithm calls the partition leaf that generates the node $v$. Then, we consider two cases: (1) $d(u', v) \geq 4 \cdot L$; (2) $d(u', v) < 4 \cdot L$, where $L = \lceil 2^{p+3} d \ln k \rceil$ used in Lemma 3.4.

In the first case, by Lemma 3.4, we have with probability at least $(1 - 1/k^3)^4 \geq 1 - 4/k^3$ (where we used Bernoulli's inequality),

$$\widetilde{D}_v \leq 2R_v \leq 2 \cdot \frac{R_{u'}}{2^4} \leq 2 \cdot \frac{4^{1/p} D_{u'}}{2^4} \leq \frac{\widetilde{D}'_u}{2}.$$

In the second case, we have $d(u, u') \geq d(u, v) - d(v, u') \geq 2^{2p+5} d \ln k$. Thus, by Lemma A.1, we have every two centers in node $u$ at distance of at least $\widetilde{D}_u/2$ are not separated at node $u'$ with probability at most

$$\left(1 - \frac{1}{d \cdot 2^{2p+2}}\right)^{2^{2p+5} d \ln k} \leq \frac{1}{k^5}.$$

By the union bound over all pairs of centers and all nodes, we have with probability at least $1 - 1/k^3$, all such pairs are separated at node $u'$. Thus, we have with probability at least $1 - 4/k^3$

$$\widetilde{D}_v \leq \widetilde{D}_{u'} = D_{u'} \leq \frac{\widetilde{D}_u}{2}.$$

$\square$

## A.2   Proof of Lemma 3.3

**Lemma 3.3.** *For some absolute constant $\beta > 0$, we have for any step $t^*$*

$$\mathbf{E}\left[\frac{\mathrm{cost}_p(x, \mathcal{T})}{\|x - \mathcal{T}_{t^*}(x)\|_p} \mid \mathcal{T}_{t^*}\right] \leq 3 + \frac{2A_k}{k} + \beta \cdot p(\log k)^{1 + \frac{1}{p} - \frac{1}{p^2}} \cdot \log(A_k \log^2 k).$$

*Proof of Lemma 3.3.* Fix an arbitrary point $x \in X$. Without loss of generality, suppose the step $t^* = 1$, in which case $\mathcal{T}_{t^*}(x) = c$ is the closest center to $x$ in $C$. Otherwise, if $t^* > 1$, then conditioned on $\mathcal{T}_{t^*}$, we consider the subinstance consisting of centers that lie in the same leaf of $\mathcal{T}_{t^*}$ as $x$.

We consider all steps in which the algorithm samples a cut to split the node containing $x$ in the partial tree. With a slight abuse of notation, we index these steps by $t = 1, 2, \ldots$. Note that some of these sampled cuts may be rejected by the algorithm if they fail to separate any centers within the node. Let $\mathcal{T}_t$ be the partially built tree before the cut at step $t$ and let $u_t$ be the node containing $x$ in $\mathcal{T}_t$. The sequence of nodes $u_1, u_2, \ldots$ thus form a path from the root to the leaf in the final tree $\mathcal{T}$ that contains $x$.[2] We divide the iterations into consecutive parts $P_1, \cdots, P_S$, each corresponding to one of the $S$ partition leaf calls. Within each part $P_s$, all steps $t \in P_s$ for $t \in P_s$ occur in the same partition leaf call and share the same anchor point $m^s$. Since the STOPPING_ORACLE ensures that for each PARTITION_LEAF call, when partitioning stops, each leaf contains at most a $\tilde{\gamma}$ fraction of the centers in its root for some constant $\tilde{\gamma} < 1$, the number of partition leaf calls is bounded by $O(\log k)$.

Suppose that at step $t$, the point $x$ and the center $c$ are contained in the same node $u_t$ before the cut is applied. Let $\omega_t = (i, \vartheta)$ be the cut selected by the algorithm at this step. We define the penalty $\phi_t(\omega_t)$, or equivalently $\phi_t(i, \vartheta)$, for the cut $(i, \vartheta)$ at step $t$ as follows. If $x$ and $c$ are not separated by cut $(i, \vartheta)$, then we set $\phi_t(i, \vartheta) = 0$. Otherwise, the penalty is given by

$$\phi_t(i, \vartheta) = \mathbf{E}[\mathrm{cost}_p(x, \mathcal{T}) \mid \mathcal{T}_t, \omega_t = (i, \vartheta)] - \|x - c\|_p.$$

We now show two upper bounds on this penalty term. Conditioned on the partial tree $\mathcal{T}_t$, we know that in the final tree $\mathcal{T}$, the point $x$ must eventually be assigned to a center in $C_{u_t}$, the set of centers

---

[2]Some of the nodes in the path may appear multiple times in the sequence since certain cuts may be rejected by the algorithm, leaving the node containing $x$ unchanged.

contained in node $u_t$. By the triangle inequality, the final cost for $x$ is at most $\|x - c\|_p + D_t$, where $D_t$ is the diameter of node $u_t$. Thus, the penalty is at most $D_t$. If $x$ and $c$ are separated by cut $(i, \vartheta)$ at iteration $t$, then we call the center $c'$ closest to $x$ in $u_{t+1}$ as the fallback center. Define $M_t(i, \vartheta) = \|x - c'\|_p$ as the distance from $x$ to its fallback center. By the definition of $A_k$, we have the penalty in this case is at most $A_k \cdot M_t(i, \vartheta)$. Combining both bounds, we obtain $\phi_t(i, \vartheta) \le \min\{D_t, A_k M_t(i, \vartheta)\}$.

Let $L = \lceil 2^{p+3} d \ln k \rceil$ and $L' = \lceil 2^{2p+6} d \ln k \rceil$. We define the stopping time $\tau$ to be the first step $t$ such that one of the following events happens: (1) $R_t < \|x - c\|_p$; (2) $x$ and $c$ are separated by the cut chosen at step $t$; (3) $\widetilde{D}_t \ge \widetilde{D}_{t-L'}/2$ for $t > L'$; (4) $R_t \ge R_{t-L}/2$ for $t > L$. We define four disjoint events as follows,

- $\mathcal{E}_1 = \{R_\tau < \|x - c\|_p\}$,

- $\mathcal{E}_2 = \{x \text{ and } c \text{ are separated by the cut chosen at step } \tau\} \setminus \mathcal{E}_1$,

- $\mathcal{E}_3 = \{\widetilde{D}_\tau \ge \widetilde{D}_{\tau-L'}/2, \tau > L'\} \setminus (\mathcal{E}_1 \cup \mathcal{E}_2)$,

- $\mathcal{E}_4 = \{R_\tau \ge R_{\tau-L}/2, \tau > L\} \setminus (\mathcal{E}_1 \cup \mathcal{E}_2 \cup \mathcal{E}_3)$.

We call $\mathcal{E}_1, \mathcal{E}_2$ good events and $\mathcal{E}_3, \mathcal{E}_4$ bad events. By Lemma 3.4 and 3.7, we have that the events $\mathcal{E}_3$ and $\mathcal{E}_4$ happen with probability at most $\Pr\{\mathcal{E}_3\} \le 1/k$ and $\Pr\{\mathcal{E}_4\} \le 1/k$. If either $\mathcal{E}_3$ or $\mathcal{E}_4$ occurs, we upper bound the expected cost of $x$ in $\mathcal{T}$ by $A_k \cdot \|x - c\|_p$ since $x$ and $c$ remain unseparated at step $\tau$. Therefore, the expected cost of point $x$ given by the tree $\mathcal{T}$ is at most

$$
\begin{aligned}
\mathbf{E}[\text{cost}_p(x, \mathcal{T})] &= \mathbf{E}[\text{cost}_p(x, \mathcal{T}) \, \mathbf{1}\{\mathcal{E}_1 \cup \mathcal{E}_2\}] + \mathbf{E}[\text{cost}_p(x, \mathcal{T}) \mid \mathcal{E}_3 \cup \mathcal{E}_4] \Pr\{\mathcal{E}_3 \cup \mathcal{E}_4\} \\
&\le \mathbf{E}[\text{cost}_p(x, \mathcal{T}) \, \mathbf{1}\{\mathcal{E}_1 \cup \mathcal{E}_2\}] + A_k \|x - c\|_p \cdot \frac{2}{k}.
\end{aligned}
$$

We then bound the expected cost of point $x$ under the good events, $\mathbf{E}[\text{cost}_p(x, \mathcal{T}) \, \mathbf{1}\{\mathcal{E}_1 \cup \mathcal{E}_2\}]$.

When the event $\mathcal{E}_1$ happens, we have $x$ and $c$ are not separated before step $\tau$. Since the diameters of nodes containing $x$ are non-increasing, the final cost for $x$ in this case can be bounded by

$$
\|x - c\|_p + D_\tau \le \|x - c\|_p + 2R_\tau < 3\|x - c\|_p.
$$

Thus, we have
$$
\mathbf{E}[\text{cost}_p(x, \mathcal{T}) \, \mathbf{1}\{\mathcal{E}_1\}] \le 3\|x - c\|_p \cdot \Pr\{\mathcal{E}_1\} \le 3\|x - c\|_p.
$$

We now turn to analyzing the event $\mathcal{E}_2$. We further partition this event based on the step at which $x$ and $c$ are first separated. For each step $t \ge 1$, we define

$$
\mathcal{E}_{2,t} = \{x \text{ and } c \text{ are separated by the cut chosen at step } \tau \, \& \, \tau = t\} \setminus \mathcal{E}_1.
$$

These events $\mathcal{E}_{2,t}$ are disjoint and we have $\mathcal{E}_2 = \bigcup_{t \ge 1} \mathcal{E}_{2,t}$. Therefore, the expected cost of $x$ under $\mathcal{E}_2$ can be expressed as

$$
\mathbf{E}[\text{cost}_p(x, \mathcal{T}) \, \mathbf{1}\{\mathcal{E}_2\}] = \sum_{t=1}^{\infty} \mathbf{E}[\text{cost}_p(x, \mathcal{T}) \, \mathbf{1}\{\mathcal{E}_{2,t}\}].
$$

We upper bound the expected cost of $x$ under event $\mathcal{E}_2$ by Lemma A.2.

By combining all events $\mathcal{E}_1, \mathcal{E}_2, \mathcal{E}_3, \mathcal{E}_4$, we have that the expected cost of $x$ is at most

$$
\begin{aligned}
\mathbf{E}[\text{cost}_p(x, \mathcal{T})] &= \mathbf{E}[\text{cost}_p(x, \mathcal{T}) \, \mathbf{1}\{\mathcal{E}_1\}] + \mathbf{E}[\text{cost}_p(x, \mathcal{T}) \, \mathbf{1}\{\mathcal{E}_2\}] + \mathbf{E}[\text{cost}_p(x, \mathcal{T}) \, \mathbf{1}\{\mathcal{E}_3 \cup \mathcal{E}_4\}] \\
&\le \left(3 + \frac{2A_k}{k}\right) \|x - c\|_p + \sum_{t=1}^{\infty} \mathbf{E}[\phi_t(\omega_t) \, \mathbf{1}\{\mathcal{E}_{2,t}\}] \\
&\le \left(3 + \frac{2A_k}{k} + \beta \cdot p(\log k)^{1 + \frac{1}{p} - \frac{1}{p^2}} \cdot \log(A_k \log^2 k)\right) \|x - c\|_p,
\end{aligned}
$$

where $\beta$ is an absolute constant. We now proceed to prove Lemma A.2. $\qquad \square$

**Lemma A.2.** *For some absolute constant $\beta > 0$, we have*

$$\sum_{t=1}^{\infty} \mathbf{E}[\phi_t(\omega_t)\,\mathbf{1}\{\mathcal{E}_{2,t}\}] \leq \beta \cdot p(\log k)^{1+\frac{1}{p}-\frac{1}{p^2}} \cdot \log(A_k \log^2 k)\|x - c\|_p.$$

*Proof.* Under the event $\mathcal{E}_2$, the point $x$ and the center $c$ are separated by a cut. We classify the cut that separates $x$ and $c$ into three cases as follows. We first recall the definitions of light and heavy steps, as well as safe and unsafe cuts, given in Definitions 3.8 and 3.9.

Fix a parameter $\alpha > 0$ which is specified later. We say that the step $t$ is a *light* step if

$$R_t \leq 6\log^\alpha k \max\{\|x - m^t\|_p, \|c - m^t\|_p\},$$

where $m^t$ is the anchor of the node $u_t$. Otherwise, we call it a *heavy* step. Furthermore, if the cut separates $x$ and $c$ at a light step, then we call it a light cut; otherwise, it is a heavy cut. Additionally, at step $t$, we say that a cut $\omega_t = (i, \vartheta)$ that separates $x$ and $c$ is *safe*, if

$$A_k M_t(i, \vartheta) < \frac{R_t}{6^p \log^2 k}.$$

Otherwise, we call this cut *unsafe*.

Then, we split the analysis into three cases: (1) safe cuts; (2) light and unsafe cuts; (3) heavy and unsafe cuts.

**Case 1 (Safe cuts):** Suppose the event $\mathcal{E}_{2,t}$ happens and $x$ and $c$ are separated by a safe cut $\omega_t = (i, \vartheta)$. By definition, a safe cut satisfies that the distance from $x$ to the fallback center $c'$ after separation is significantly smaller than the current radius, specifically $A_k M_t(i, \vartheta) < R_t/(6^p \log^2 k)$. In this case, we use $A_k M_t(i, \vartheta)$ as an upper bound on the penalty incurred by separating $x$ and $c$.

For each step $t$, coordinate $i \in \{1, 2, \cdots, d\}$, and direction $\sigma \in \{-1, 1\}$, we define the safe cut set

$$G_{t,i,\sigma} = \left\{\theta : A_k M_t(i, m_i^t + \sigma\theta) < \frac{R_t}{6^p \log^2 k}\ \&\ (i, m_i^t + \sigma\theta) \text{ separates } x \text{ and } c\right\},$$

which contains all parameters $\theta \in \mathbb{R}$ such that the corresponding cut $\omega_t = (i, m_i^t + \sigma\theta)$ is safe. Then, the expected penalty due to safe cuts is at most

$$\sum_{t=1}^{\infty} \mathbf{E}\left[\phi_t(\omega_t)\,\mathbf{1}\{\omega_t \text{ is safe}\}\,\mathbf{1}\{\mathcal{E}_{2,t}\}\right]$$

$$\leq \sum_{t=1}^{\infty} \mathbf{E}\left[A_k M_t(i_t, m_i^t + \sigma_t\theta_t)\,\mathbf{1}\{\theta_t \in G_{t,i_t,\sigma_t}\}\,\mathbf{1}\{\mathcal{E}_{2,t}\}\right]$$

$$\leq \sum_{t=1}^{\infty} \sum_{i=1}^{d} \sum_{\sigma \in \{-1,1\}} \frac{1}{2d} \int_{G_{t,i,\sigma}} A_k M_t(i, m_i^t + \sigma\theta) \cdot \frac{p \cdot \theta^{p-1}}{R_t^p} \cdot \mathbf{1}\{\mathcal{E}_{2,t}\} \cdot \mathrm{d}\theta$$

$$= \sum_{t=1}^{\infty} \sum_{i=1}^{d} \sum_{\sigma \in \{-1,1\}} \frac{1}{2d} \int_{G_{t,i,\sigma}} \frac{A_k M_t(i, m_i^t + \sigma\theta)}{R_t} \cdot \frac{p \cdot \theta^{p-1}}{R_t^{p-1}} \cdot \mathbf{1}\{\mathcal{E}_{2,t}\} \cdot \mathrm{d}\theta.$$

Here, the second inequality uses the fact that the coordinate $i$ is chosen uniformly from $\{1, 2, \cdots, d\}$ and the direction $\sigma$ is chosen uniformly from $\{-1, 1\}$ and that $\theta$ is drawn from a distribution with density $p\theta^{p-1}/R_t^p$. The safe cuts are those with $\theta \in G_{t,i,\sigma}$.

Now we derive an upper bound for $\theta/R_t$ to control the integral. Since center $c$ lies in node $u_t$, we have $\|c - m^t\|_p \leq R_t$. Additionally, since the event $\mathcal{E}_1$ does not occur, we have $R_t \geq \|x - c\|_p$. Using the triangle inequality, we have

$$\|x - m^t\|_p \leq \|x - c\|_p + \|c - m^t\|_p \leq 2R_t.$$

Therefore, we have

$$3R_t \geq \|x - m^t\|_p + \|c - m^t\|_p.$$

Furthermore, for any $\theta \in G_{t,i,\sigma}$, the cut $(i, m_i^t + \sigma\theta)$ separates $x$ and $c$, which implies

$$\theta \leq \max\{|x_i - m_i^t|, |c_i - m_i^t|\}.$$

Therefore, conditioned on the event $\mathbf{1}\{\mathcal{E}_{2,t}\} = 1$, we have for any $\theta \in G_{t,i,\sigma}$,

$$\frac{\theta}{R_t} \leq \frac{3 \max\{|x_i - m_i^t|, |c_i - m_i^t|\}}{\|x - m^t\|_p + \|c - m^t\|_p}.$$

We now analyze each partition leaf call separately. Fix a partition leaf call $P_s$. Throughout this partition leaf call, the anchor $m^s$ stays the same. Thus, the expected penalty due to safe cuts within this call is at most

$$\sum_{t \in P_s} \mathbf{E}\left[\phi_t(\omega_t)\,\mathbf{1}\{\omega_t \text{ is safe}\}\,\mathbf{1}\{\mathcal{E}_{2,t}\}\right]$$

$$\leq \frac{p}{2d} \sum_{i=1}^{d} \frac{3^{p-1} \max\{|x_i - m_i^s|, |c_i - m_i^s|\}^{p-1}}{(\|x - m^s\|_p + \|c - m^s\|_p)^{p-1}} \sum_{t \in P_s} \sum_{\sigma \in \{-1,1\}} \int_{G_{t,i,\sigma}} \frac{A_k M_t(i, m_i^s + \sigma\theta)}{R_t} \cdot d\theta.$$

By Hölder's inequality, the expected penalty above is at most

$$\frac{p}{2d} \cdot \left(\sum_{i=1}^{d} \frac{3^p \max\{|x_i - m_i^s|, |c_i - m_i^s|\}^p}{(\|x - m^s\|_p + \|c - m^s\|_p)^p}\right)^{\frac{p-1}{p}}$$

$$\cdot \left(\sum_{i=1}^{d} \left(\sum_{t \in P_s} \sum_{\sigma \in \{-1,1\}} \int_{G_{t,i,\sigma}} \frac{A_k M_t(i, m_i^s + \sigma\theta)}{R_t} \cdot d\theta\right)^p\right)^{\frac{1}{p}}.$$

Then, we bound the two terms in the above formula separately. First, we have

$$\sum_{i=1}^{d} \max\{|x_i - m_i^s|, |c_i - m_i^s|\}^p \leq \|x - m^s\|_p^p + \|c - m^s\|_p^p.$$

Thus, we have the first term is

$$\left(\sum_{i=1}^{d} \frac{3^p \max\{|x_i - m_i^s|, |c_i - m_i^s|\}^p}{(\|x - m^s\|_p + \|c - m^s\|_p)^p}\right)^{\frac{p-1}{p}} \leq 3^{p-1}.$$

We now bound the second term. Note that for any fixed cut $\omega = (i, \vartheta)$, the fallback distance $M_t(i, \vartheta)$ is non-decreasing with respect to the step $t$. Meanwhile, within each partition leaf call $P_s$, the radius $R_t$ is non-increasing and decreases by a factor of 2 after every $L$ steps under event $\mathcal{E}_2$. Therefore, for each coordinate $i \in \{1, 2, \cdots, d\}$, we have

$$\sum_{t \in P_s} \sum_{\sigma \in \{-1,1\}} \int_{G_{t,i,\sigma}} \frac{A_k M_t(i, m_i^s + \sigma\theta)}{R_t} \cdot d\theta$$

$$\leq \int \sum_{t \in P_s} \sum_{\sigma \in \{-1,1\}} \frac{A_k M_t((i, m_i^s + \sigma\theta))}{R_t} \mathbf{1}\{\theta \in G_{t,i,\sigma}\} \cdot d\theta$$

$$\leq 4L \cdot \frac{1}{6^p \log^2 k} \cdot |x_i - c_i|,$$

where the last inequality follows from the definition of safe cuts, which ensures that $A_k M_t(i, \vartheta) < \frac{R_t}{6^p \log^2 k}$ whenever $\theta \in G_{t,i,\sigma}$, and $\frac{A_k M_t(i, \vartheta)}{R_t}$ forms a geometric sequence increases by a factor of 2 every $L$ steps. Therefore, we have the second term is at most

$$\left(\sum_{i=1}^{d} \left(\sum_{t \in P_s} \sum_{\sigma \in \{-1,1\}} \int_{G_{t,i,\sigma}} \frac{A_k M_t(i, m_i^s + \sigma\theta)}{R_t} \cdot d\theta\right)^p\right)^{\frac{1}{p}} \leq 4L \cdot \frac{1}{6^p \log^2 k} \cdot \|x - c\|_p.$$

Since there are at most $O(\log k)$ partition leaf calls and $L = \lceil 2^{p+3} d \ln k \rceil$, the expected penalty due to safe cuts is at most

$$O(\log k) \cdot \frac{p}{2d} \cdot 3^{p-1} \cdot 4L \cdot \frac{1}{6^p \log^2 k} \|x - c\|_p \le O(p) \cdot \|x - c\|_p.$$

**Case 2 (Light and unsafe cuts):** In this case, we have that the radius $R_t$ is relatively small compared to $\|x - m^t\|_p$ and $\|c - m^t\|_p$, specifically, $R_t \le 6 \log^\alpha k \max\{\|x - m^t\|_p, \|c - m^t\|_p\}$. Therefore, in this case, we use $D_t \le 2R_t$ as an upper bound on the penalty. Then, the expected penalty due to a light and unsafe cut is

$$\sum_{t=1}^\infty \mathbf{E}\left[\phi_t(\omega_t) \mathbf{1}\{t \text{ is light}\} \mathbf{1}\{\omega_t \text{ is unsafe}\} \mathbf{1}\{\mathcal{E}_{2,t}\}\right] \le \sum_{t=1}^\infty \mathbf{E}\left[\phi_t(\omega_t) \mathbf{1}\{t \text{ is light}\} \mathbf{1}\{\mathcal{E}_{2,t}\}\right]$$

$$\le \sum_{t=1}^\infty \mathbf{E}\left[2R_t \mathbf{1}\{t \text{ is light}\} \mathbf{1}\{\mathcal{E}_{2,t}\}\right].$$

For each step $t$, suppose both $x$ and $c$ are contained in the node $u_t$. We define the new event $\mathcal{E}'_t$ as the event that either $x$ or $c$ is first separated from the anchor $m^t$ by the cut chosen at step $t$. To bound the expected penalty above, we show that

$$\sum_{t=1}^\infty \mathbf{E}\left[R_t \mathbf{1}\{t \text{ is light}\} \mathbf{1}\{\mathcal{E}_{2,t}\}\right] \le 24p \log^\alpha k \cdot \|x - c\|_p \cdot \sum_{t=1}^\infty \mathbf{E}\left[\mathbf{1}\{t \text{ is light}\} \mathbf{1}\{\mathcal{E}'_t\}\right].$$

To show this, we define the stochastic process $\{Y_t\}_{t \ge 0}$ as follows. Let $Y_0 = 0$ and for any $t \ge 1$,

$$Y_t = \sum_{t'=1}^t \left(R_{t'} \mathbf{1}\{\mathcal{E}_{2,t'}\} - \|x - c\|_p \cdot 24p \log^\alpha k \mathbf{1}\{\mathcal{E}'_{t'}\}\right) \mathbf{1}\{t' \text{ is light}\}.$$

We now show that this stochastic process $\{Y_t\}_{t \ge 0}$ forms a supermartingale. Note that for each step $t \ge 1$, we have

$$Y_t = Y_{t-1} + \left(R_t \mathbf{1}\{\mathcal{E}_{2,t}\} - \|x - c\|_p \cdot 24p \log^\alpha k \mathbf{1}\{\mathcal{E}'_t\}\right) \mathbf{1}\{t \text{ is light}\}.$$

If step $t$ is heavy, then $Y_t = Y_{t-1}$. In the following analysis, we focus on the case where $t$ is a light step and both $x$ and $c$ are contained in the node $u_t$. In this case, we first analyze the probability that the chosen cut separates $x$ and $c$, and the probability that separates either $x$ or $c$ from the anchor $m^t$.

**Claim A.3.** *Suppose both $x$ and $c$ are contained in the node $u_t$ at this step $t$. Then, the probability that $x$ and $c$ are separated by the chosen cut is at most*

$$\Pr\{x \text{ and } c \text{ are separated at step } t \mid \mathcal{T}_t\} \le \frac{p}{d} \|x - c\|_p \frac{\|x - m^t\|_p^{p-1} + \|c - m^t\|_p^{p-1}}{R_t^p}.$$

*The probability that either $x$ or $c$ is first separated from $m^t$ by the cut chosen at step $t$ is at least*

$$\Pr\{\mathcal{E}'_t \mid \mathcal{T}_t\} \ge \frac{1}{2d} \cdot \frac{\max\{\|x - m^t\|_p^p, \|c - m^t\|_p^p\}}{R_t^p}.$$

Thus, we have for a light step $t$,

$$\mathbf{E}[Y_t \mid \mathcal{T}_t] - Y_{t-1} = R_t \Pr\{\mathcal{E}_{2,t} \mid \mathcal{T}_t\} - 24p \log^\alpha k \cdot \|x - c\|_p \Pr\{\mathcal{E}'_t \mid \mathcal{T}_t\}$$

$$\le \frac{p}{d} \|x - c\|_p \frac{\|x - m^t\|_p^{p-1} + \|c - m^t\|_p^{p-1}}{R_t^{p-1}} - \|x - c\|_p \frac{12p \log^\alpha k}{d} \frac{\max\{\|x - m^t\|_p^p, \|c - m^t\|_p^p\}}{R_t^p}$$

$$\le \frac{p}{d} \|x - c\|_p \frac{2 \max\{\|x - m^t\|_p^{p-1}, \|c - m^t\|_p^{p-1}\}}{R_t^{p-1}} \left(1 - 6 \log^\alpha k \frac{\max\{\|x - m^t\|, \|c - m^t\|\}}{R_t}\right)$$

$$\le 0,$$

where the last inequality follows from the definition of a light step. Therefore, $\{Y_t\}_{t \geq 0}$ is a supermartingale. Hence, $\mathbf{E}[Y_T] \leq \mathbf{E}[Y_0]$ for every fixed $T$. Since $\mathbf{E}[Y_0] = 0$, we have $\mathbf{E}[Y_T] \leq 0$ and

$$\sum_{t=1}^{T} \mathbf{E}\left[R_t\, \mathbf{1}\{t \text{ is light}\}\, \mathbf{1}\{\mathcal{E}_{2,t}\}\right] \leq 24p \log^\alpha k \cdot \|x - c\|_p \cdot \sum_{t=1}^{T} \mathbf{E}\left[\mathbf{1}\{t \text{ is light}\}\, \mathbf{1}\{\mathcal{E}'_t\}\right].$$

Letting $T \to \infty$, we obtain

$$\sum_{t=1}^{\infty} \mathbf{E}\left[R_t\, \mathbf{1}\{t \text{ is light}\}\, \mathbf{1}\{\mathcal{E}_{2,t}\}\right] \leq 24p \log^\alpha k \cdot \|x - c\|_p \cdot \sum_{t=1}^{\infty} \mathbf{E}\left[\mathbf{1}\{t \text{ is light}\}\, \mathbf{1}\{\mathcal{E}'_t\}\right].$$

To bound the right-hand side, it suffices to control the expected number of times the event $\mathcal{E}'_t$ occurs. Recall that $\mathcal{E}'_t$ denotes the event that either $x$ or $c$ is first separated from the anchor $m^t$ at step $t$.

We begin by noting that the number of partition-leaf calls is at most $O(\log k)$. Within each partition-leaf call, the anchor point $m^t$ remains fixed, and once $x$ is separated from $m^t$, it will no longer be involved in further cuts associated with that anchor. Therefore, $x$ can be separated from $m^t$ in at most one step per partition-leaf call, contributing at most $O(\log k)$ occurrences of $\mathcal{E}'_t$. Additionally, observe that the center $c$ can be separated from the anchor $m^t$ without $x$ being separated at most once. After such a separation, $c$ will no longer lie in the same node as $x$ and will not contribute to future events $\mathcal{E}'_t$.

Combining these observations, we conclude that the expected number of steps where $\mathcal{E}'_t$ occurs is at most $O(\log k)$, which yields

$$\mathbf{E}\left[\sum_{t=1}^{\infty} \phi_t(\omega_t)\, \mathbf{1}\{t \text{ is light}\}\, \mathbf{1}\{\omega_t \text{ is unsafe}\}\, \mathbf{1}\{\mathcal{E}_{2,t}\}\right] \leq O\left(p \log^{1+\alpha} k\right) \|x - c\|_p.$$

**Case 3 (Heavy and unsafe cuts):** Suppose the event $\mathcal{E}_{2,t}$ occurs and that $x$ and $c$ are separated by an unsafe cut $\omega_t = (i, \vartheta)$. For each step $t$, coordinate $i \in \{1, 2, \cdots, d\}$, and direction $\sigma \in \{-1, 1\}$, we define the the corresponding unsafe cut set as

$$U_{t,i,\sigma} = \left\{\theta : A_k M_t(i, m_i^t + \sigma\theta) \geq \frac{R_t}{6p \log^2 k} \ \& \ (i, m_i^t + \sigma\theta) \text{ separates } x \text{ and } c\right\},$$

that is, the set of threshold $\theta$ for which the cut $(i, m_i^t + \sigma\theta)$ is both unsafe and separates $x$ from $c$. Let $\delta_{i,\sigma}(t) = \mu(U_{t,i,\sigma})$ denote the Lebesgue measure of the set $U_{t,i,\sigma}$ and define $\delta_i(t) = \delta_{i,-1}(t) + \delta_{i,1}(t)$ as the total measure across both directions for coordinate $i$.

Thus, the probability that $\omega_t$ is an unsafe cut is at most

$$\begin{aligned}
\Pr\{\omega_t \text{ is unsafe}\} &= \frac{1}{2d} \int_{U_{t,i,\sigma}} \frac{p \cdot \theta^{p-1}}{R_t^p} \cdot \mathrm{d}\theta \\
&\leq \frac{p}{2d} \sum_{i=1}^{d} \sum_{\sigma \in \{-1,1\}} \frac{\max\{|x_i - m_i^t|, |c_i - m_i^t|\}^{p-1}}{R_t^p} \cdot \delta_{i,\sigma}(t) \\
&= \frac{p}{2d} \sum_{i=1}^{d} \frac{\max\{|x_i - m_i^t|, |c_i - m_i^t|\}^{p-1}}{R_t^p} \cdot \delta_i(t).
\end{aligned}$$

In this case, we use the radius $2R_t$ as the upper bound on the penalty for separating $x$ and $c$. Therefore, the expected penalty incurred from heavy and unsafe cuts is bounded by

$$\begin{aligned}
&\sum_{t=1}^{\infty} \mathbf{E}\left[\phi_t(\omega_t)\, \mathbf{1}\{\omega_t \text{ is unsafe}\}\, \mathbf{1}\{t \text{ is heavy}\}\, \mathbf{1}\{\mathcal{E}_{2,t}\}\right] \\
&\leq \sum_{t=1}^{\infty} \mathbf{E}\left[2R_t \Pr\{\omega_t \text{ is unsafe}\}\, \mathbf{1}\{t \text{ is heavy}\}\, \mathbf{1}\{\mathcal{E}_{2,t}\}\right] \\
&\leq \sum_{t=1}^{\infty} \mathbf{E}\left[R_t \sum_{i=1}^{d} \frac{p}{d} \delta_i(t) \frac{\max\{|x_i - m_i^t|, |c_i - m_i^t|\}^{p-1}}{R_t^p}\, \mathbf{1}\{t \text{ is heavy}\}\, \mathbf{1}\{\mathcal{E}_{2,t}\}\right].
\end{aligned}$$

Since step $t$ is heavy, we have $R_t \geq 6 \log^\alpha k \cdot \max\{\|x - m^t\|_p, \|c - m^t\|_p\}$, which implies

$$\frac{1}{R_t^{p-1}} \mathbf{1}\{t \text{ is heavy}\} \leq \frac{1}{(6 \log^\alpha k)^{p-1} \max\{\|x - m_t\|_p, \|c - m_t\|_p\}^{p-1}}.$$

Substituting this into the previous bound, we obtain that the expected penalty in this case is at most

$$\sum_{t=1}^{\infty} \mathbf{E}\left[\phi_t(\omega_t) \mathbf{1}\{\omega_t \text{ is unsafe}\} \mathbf{1}\{t \text{ is heavy}\} \mathbf{1}\{\mathcal{E}_{2,t}\}\right]$$

$$\leq \sum_{t=1}^{\infty} \mathbf{E}\left[\sum_{i=1}^{d} \frac{p}{d} \delta_i(t) \frac{\max\{|x_i - m_i^t|, |c_i - m_i^t|\}^{p-1}}{(6 \log^\alpha k)^{p-1} \max\{\|x - m_t\|_p, \|c - m_t\|_p\}^{p-1}} \mathbf{1}\{\mathcal{E}_{2,t}\}\right].$$

Note that all steps within the same partition leaf call $P_s$ share the same anchor point. Let $\bar{m}^s$ denote the anchor point used in the partition leaf call $P_s$, and define $\Delta_i(s) = \sum_{t \in P_s} \delta_i(t)$. Then, the expected penalty above is at most

$$\mathbf{E}\left[\sum_{s=1}^{S} \sum_{t \in P_s} \sum_{i=1}^{d} \frac{p}{d} \delta_i(t) \frac{\max\{|x_i - \bar{m}_i^s|, |c_i - \bar{m}_i^s|\}^{p-1}}{(6 \log^\alpha k)^{p-1} \max\{\|x - \bar{m}^s\|_p, \|c - \bar{m}^s\|_p\}^{p-1}} \mathbf{1}\{\mathcal{E}_{2,t}\}\right]$$

$$\leq \mathbf{E}\left[\sum_{s=1}^{S} \frac{p}{d} \sum_{i=1}^{d} \Delta_i(s) \frac{\max\{|x_i - \bar{m}_i^s|, |c_i - \bar{m}_i^s|\}^{p-1}}{(6 \log^\alpha k)^{p-1} \max\{\|x - \bar{m}^s\|_p, \|c - \bar{m}^s\|_p\}^{p-1}} \mathbf{1}\{\mathcal{E}_{2,t}\}\right].$$

Let $\Delta(s)$ denote the $d$-dimensional vector with coordinates $\Delta_i(s)$ for $i \in \{1, 2, \cdots, d\}$. Applying Hölder's inequality, we get

$$\sum_{t=1}^{\infty} \mathbf{E}\left[\phi_t(\omega_t) \mathbf{1}\{\omega_t \text{ is unsafe}\} \mathbf{1}\{t \text{ is heavy}\} \mathbf{1}\{\mathcal{E}_{2,t}\}\right]$$

$$\leq \mathbf{E}\left[\sum_{s=1}^{S} \frac{p}{d} \|\Delta(s)\|_p \frac{\left(\sum_{i=1}^{d} |x_i - \bar{m}_i^s|^p\right)^{\frac{p-1}{p}} + \left(\sum_{i=1}^{d} |c_i - \bar{m}_i^s|^p\right)^{\frac{p-1}{p}}}{(6 \log^\alpha k)^{p-1} \max\{\|x - \bar{m}^s\|_p, \|c - \bar{m}^s\|_p\}^{p-1}} \mathbf{1}\{\mathcal{E}_2\}\right]$$

$$\leq \mathbf{E}\left[\sum_{s=1}^{S} \frac{p}{d} \|\Delta(s)\|_p \frac{\|x - \bar{m}^s\|_p^{p-1} + \|c - \bar{m}^s\|_p^{p-1}}{(6 \log^\alpha k)^{p-1} \max\{\|x - \bar{m}^s\|_p, \|c - \bar{m}^s\|_p\}^{p-1}} \mathbf{1}\{\mathcal{E}_2\}\right]$$

$$\leq \frac{p}{(6 \log^\alpha k)^{p-1} d} \mathbf{E}\left[\sum_{s=1}^{S} \|\Delta(s)\|_p \cdot \mathbf{1}\{\mathcal{E}_2\}\right].$$

Finally, we use the following claim to bound the expected penalty.

**Claim A.4.** *We have*

$$\mathbf{E}\left[\sum_{s=1}^{S} \|\Delta(s)\|_p \cdot \mathbf{1}\{\mathcal{E}_2\}\right] = O\left(4^p \cdot d \cdot \log^{2 - \frac{1}{p}} k \cdot \log(6^p \cdot A_k \cdot \log^2 k)\right) \|x - c\|_p.$$

By Claim A.4, we have that the expected penalty in this case is at most

$$\frac{p}{(6 \log^\alpha k)^{p-1} d} \mathbf{E}\left[\sum_{s=1}^{S} \|\Delta(s)\|_p \cdot \mathbf{1}\{\mathcal{E}_{2,t}\}\right]$$

$$\leq \frac{p}{(6 \log^\alpha k)^{p-1} d} \cdot O\left(4^p \cdot d \cdot \log^{2 - \frac{1}{p}} k \cdot \log(6^p \cdot A_k \cdot \log^2 k)\right) \|x - c\|_p$$

$$\leq O\left((\log k)^{2 - \frac{1}{p} - \alpha(p-1)} \cdot \log(A_k \cdot \log^2 k)\right) \|x - c\|_p.$$

Combining all three cases and setting $\alpha = 1/p - 1/p^2$ we get the conclusion.

To complete the proof, we prove Claim A.3 and A.4 below. $\qquad\square$

*Proof of Claim A.3.* We first analyze the probability that $x$ and $c$ are separated by the cut chosen at step $t$. To bound the separation probability, we fix a coordinate $i \in \{1, 2, \cdots, d\}$ and consider the probability that the cut on coordinate $i$ separates $x$ and $c$.

Suppose $x$ and $c$ are on the same side of anchor $m^t$ in coordinate $i$. Then, the threshold cut $\omega_t = (i, m_i^t + \sigma\theta)$ separates $x$ and $c$ if and only if $\sigma$ has the same sign as $x_i - m_i^t$ and $\theta$ is between $|x_i - m_i^t|$ and $|c_i - m_i^t|$. Thus, the separation probability on this coordinate is at most

$$\frac{1}{2} \cdot \frac{||c_i - m_i^t|^p - |x_i - m_i^t|^p|}{R_t^p} \leq \frac{p \cdot \max\{|x_i - m_i^t|^{p-1}, |c_i - m_i^t|^{p-1}\}}{R_t^p} \cdot |x_i - c_i|,$$

where the inequality is from the mean value theorem.

Suppose $x$ and $c$ are on the opposite side of anchor $m^t$ in coordinate $i$. Then, the separation probability on this coordinate is at most

$$\frac{1}{2} \cdot \frac{|c_i - m_i^t|^p + |x_i - m_i^t|^p}{R_t^p} \leq \frac{p \cdot \max\{|x_i - m_i^t|^{p-1}, |c_i - m_i^t|^{p-1}\}}{R_t^p} \cdot |x_i - c_i|.$$

Combining all coordinates and applying Hölder's inequality, we obtain

$$\frac{1}{d} \sum_{i=1}^d \frac{p \cdot \max\{|x_i - m_i^t|^{p-1}, |c_i - m_i^t|^{p-1}\}}{R_t^p} \cdot |x_i - c_i|$$

$$\leq \frac{p}{d \cdot R_t^p} \|x - c\|_p \cdot \left( \left( \sum_{i=1}^d |x_i - m_i^t|^p \right)^{\frac{p-1}{p}} + \left( \sum_{i=1}^d |c_i - m_i^t|^p \right)^{\frac{p-1}{p}} \right)$$

$$\leq \frac{p}{d} \|x - c\|_p \cdot \frac{\|x - m^t\|_p^{p-1} + \|c - m^t\|_p^{p-1}}{R_t^p}.$$

For point $x$, the probability that it is separated from $m^t$ at step $t$ is given by

$$\frac{1}{2d} \sum_{i=1}^d \frac{|x_i - m_i^t|^p}{R_t^p} = \frac{1}{2d} \cdot \frac{\|x - m^t\|_p^p}{R_t^p}.$$

An identical argument applies to the center $c$, yielding the same expression with $\|c - m^t\|_p^p$. Therefore, the probability that either $x$ or $c$ is separated from $m^t$ by the threshold cut at step $t$ is at least

$$\frac{1}{2d} \cdot \frac{\max\{\|x - m^t\|_p^p, \|c - m^t\|_p^p\}}{R_t^p},$$

as claimed. $\square$

To prove Claim A.4, we first show the following lemma.

**Lemma A.5.** *For $k$ vectors $v^1, \cdots, v^k \in \mathbb{R}^d$ that are entrywise non-negative, we have*

$$\sum_{i=1}^k \|v^i\|_p \leq k^{1-\frac{1}{p}} \cdot \left\| \sum_{i=1}^k v^i \right\|_p.$$

*Proof.* We first upper bound the left-hand side. By Hölder's inequality, we have

$$\sum_{i=1}^k \|v^i\|_p = \sum_{i=1}^k 1 \cdot \|v^i\|_p \leq k^{\frac{1}{q}} \left( \sum_{i=1}^k \|v^i\|_p^p \right)^{\frac{1}{p}} = k^{1-\frac{1}{p}} \left( \sum_{i=1}^k \|v^i\|_p^p \right)^{\frac{1}{p}}.$$

We then lower bound the right-hand side. Since vectors $v^1, \cdots, v^k$ are nonnegative in every coordinate, we have for any coordinate $j$,

$$\left( \sum_{i=1}^k v_j^i \right)^p \geq \sum_{i=1}^k (v_j^i)^p.$$

Combining all coordinates, we have

$$\left\| \sum_{i=1}^{k} v_i \right\|_p^p = \sum_{j=1}^{d} \left( \sum_{i=1}^{k} v_j^i \right)^p \geq \sum_{j=1}^{d} \sum_{i=1}^{k} (v_j^i)^p = \sum_{i=1}^{k} \|v_i\|_p^p.$$

Combining the two parts, we get the conclusion. $\qquad\square$

*Proof of Claim A.4.* By Lemma A.5 and the number of partition leaf calls is at most $O(\log k)$, we have

$$\mathbf{E}\left[ \sum_{s=1}^{S} \|\Delta(s)\|_p \cdot \mathbf{1}\{\mathcal{E}_2\} \right] \leq O\left( \log^{1-\frac{1}{p}} k \right) \mathbf{E}\left[ \left\| \sum_{s=1}^{S} \Delta(s) \right\|_p \cdot \mathbf{1}\{\mathcal{E}_2\} \right].$$

For any fixed coordinate $i$, we have

$$\sum_{s=1}^{S} \Delta_i(s) = \sum_{s=1}^{S} \sum_{t \in P_s} \delta_i(t) = \sum_{t=1}^{\infty} \delta_i(t) = \sum_{t=1}^{\infty} \int \mathbf{1}\{\theta \in U_{t,i,1}\}\mathrm{d}\theta + \int \mathbf{1}\{\theta \in U_{t,i,-1}\}\mathrm{d}\theta.$$

We now show that each cut $\omega = (i, \vartheta)$ that separates $x$ and $c$ is unsafe in at most $L'' = L' \cdot \log(6^p \cdot A_k \cdot \log^2 k)$ steps. Consider any cut $\omega = (i, \vartheta)$ that separates $x$ and $c$. This cut $\omega$ is unsafe at step $t$ if and only if $R_t \leq 6^p \log^2 k \cdot A_k M_t(i, \vartheta)$. For every step $t$, by the triangle inequality, the penalty to the fallback center is at most $M_t(i, \vartheta) \leq D_t \leq \widetilde{D}_t$. We know that $M_t(i, \vartheta)$ is non-decreasing as $t$ increases. Let $t_\omega$ be the first step when $\omega$ is unsafe. Let $t'_\omega$ be the last step when $\omega$ is unsafe. Then, by the definition of unsafe cut, we have $R_{t_\omega} \leq 6^p \log^2 k \cdot A_k M_{t_\omega}(i, \vartheta)$. Then, we have

$$\widetilde{D}_{t'_\omega} \geq M_{t'_\omega}(\omega) \geq M_{t_\omega}(\omega) \geq \frac{R_{t_\omega}}{6^p \cdot \log^2 k \cdot A_k}.$$

Since $R_t/4^{1/p} \leq \widetilde{D}_t \leq 2R_t$, we have

$$\widetilde{D}_{t'_\omega} \geq \frac{R_{t_\omega}}{6^p \cdot \log^2 k \cdot A_k} \geq \frac{\widetilde{D}_{t_\omega}}{2 \cdot 6^p \cdot \log^2 k \cdot A_k}.$$

By Lemma 3.7, we have $\widetilde{D}_t$ decreases by a factor of 2 after $L' = \lceil 2^{2p+6}d \ln k \rceil$ steps. Thus, we have that the number of unsafe steps is at most

$$t'_\omega - t_\omega \leq L' \cdot \log_2(2 \cdot 6^p \cdot \log^2 k \cdot A_k) \leq O(4^p \cdot d \cdot \log k \cdot p \log(A_k \cdot \log^2 k)).$$

Therefore, we have that when the event $\mathcal{E}_2$ happens,

$$\sum_{s=1}^{S} \Delta_i(s) \leq O(4^p \cdot d \cdot \log k \cdot \log(6^p \cdot A_k \cdot \log^2 k))|x_i - c_i|.$$

Hence, combining all coordinates, we have

$$\mathbf{E}\left[ \left\| \sum_{s=1}^{S} \Delta^u(s) \right\|_p \mathbf{1}\{\mathcal{E}_2\} \right] = O(4^p \cdot d \cdot \log k \cdot \log(6^p \cdot A_k \cdot \log^2 k))\|x - c\|_p,$$

which completes the proof. $\qquad\square$

# B  Dynamic algorithm implementation and analysis

In this section, we provide the full description of the dynamic algorithm, along with an analysis of its approximation guarantee, update time, and recourse.

## B.1 Dynamic algorithm and approximation guarantee

We begin by presenting the detailed dynamic algorithm and proving that, after each update, the distribution of its output is equivalent to that of a corresponding static algorithm.

**Lemma B.1.** *Given a sequence of $k$ requests, where each request is either an insertion or a deletion of a single center, let $\mathcal{T}_t$ be the threshold tree maintained by the dynamic algorithm for the center set $C_t$. Let $\mathcal{T}_t'$ be the tree constructed by the static algorithm* PARTITION_LEAF *(Figure 1) with specific oracles on centers $C_t$. Then, the two trees are identically distributed $\mathcal{T}_t \stackrel{d}{=} \mathcal{T}_t'$.*

The following corollary is immediate from Lemma B.1 and Theorem 3.1.

**Corollary B.2.** *Given a sequence of requests, where each request is either an insertion or a deletion of a single center, the dynamic algorithm provides a threshold tree $\mathcal{T}_t$ for each center set $C_t$ such that for any set of points $X$,*

$$\mathbf{E}[\text{cost}_p(X, \mathcal{T}_t)] = O\left(p(\log k_t)^{1 + \frac{1}{p} - \frac{1}{p^2}} \log \log k_t\right) \text{cost}_p(X, C_t),$$

*where $k_t = |C_t|$.*

We provide a dynamic implementation of the PARTITION_LEAF procedure in Figure 2, which is applied recursively to obtain a fully dynamic version of the entire clustering algorithm. The dynamic variant of PARTITION_LEAF supports three operations: (1) REBUILD, (2) INSERT CENTER, and (3) DELETE CENTER.

We begin with the REBUILD operation, which reconstructs the subtree from scratch using the PARTITION_LEAF procedure as follows.

REBUILD: Reconstruct the subtree rooted at node $u$, partitioning all centers in $C_u$ into distinct leaves via recursive calls to the PARTITION_LEAF procedure. During each such PARTITION_LEAF call on node $v$ in this operation, the following oracle outputs are used and remain fixed throughout subsequent updates until the next rebuild:

- GET_ANCHOR sets the anchor $m^v$ as the coordinate-wise median of centers in $C_v$.
- STOPPING_ORACLE determines whether to stop accepting further cuts based on a stopping time $\rho^v$. It returns True if and only if the the timestamp $\rho$ of the input cut $\omega$ satisfies $\rho > \rho^v$. The stopping time $\rho^v$ is defined during the rebuild as the timestamp of the last accepted cut such that the main part $v_0$ contains at most half of centers in $C_v$, i.e. $|C_{v_0}| \leq |C_v|/2$.

We now describe the condition under which the rebuild operation is triggered in the dynamic algorithm. Let $u$ be the node on which this operation is applied. Suppose a center $c$ is inserted into or deleted from the set of centers assigned to $u$. For each partition leaf call, we maintain a counter that tracks the number of such updates since the last rebuild. Let $k'$ be the number of centers in node $u$ at the time of the last rebuild. When the update count exceeds $k'/4$, we rebuild the partial tree rooted at node $u$.

We now proceed to handle the update.

INSERT CENTER: Suppose a new center $c$ is inserted in the subtree rooted at a node $u$. The algorithm calls GET_EARLIEST_CUT to find the earliest cut $\omega$ in the pre-generated sequence with its arrival time $\rho$ that separates $c$ from the anchor $m^u$. Let $(\omega_1', \rho_1'), \cdots, (\omega_r', \rho_r')$ be the cuts currently used in this partition leaf call. Let $\rho^u$ be the stopping time assigned to this partition leaf call during its most recent rebuild. We consider three cases as follows: (1) $\rho = \rho_j'$ for some $j \in [r]$; (2) $\rho > \rho^u$; (3) $\rho \leq \rho^u$ and $\rho \neq \rho_j'$ for any $j \in [r]$.

Case (1): Assign this new center $c$ to the node $v$ generated by cut $\omega_j'$ and recursively maintain the partition leaf call rooted at $v$.

Case (2): This new center $c$ remains in the main part $u_0$ until this partition leaf call ends. We then recursively maintain the partition leaf call on the main part $u_0$.

Case (3): It finds the smallest index $j \in [r]$ such that $\rho < \rho_j'$ or sets $j = r + 1$ if no such index exists. Then we insert this new cut $\omega$ at position $j$ and add a new leaf node containing $c$ to the tree.

DELETE CENTER: Now suppose a center $c \in C_u$ is deleted. We locate the leaf node containing $c$ in this partition leaf call. If this leaf contains only one center $c$, we remove both the leaf and the cut that created it. Otherwise, we delete $c$ from the leaf and maintain the next partition call recursively.

**Algorithm** DYNAMIC_PARTITION_LEAF

**Input:** A sequence of updates $Q = (q_1, q_2, \dots)$, where each $q_t$ is either: INSERT$(c)$: insert a new center $c \in \mathbb{R}^d$; or DELETE$(c)$: delete a center $c$.

**Output:** For each update $q_t$ in $Q$, maintain a threshold tree $\mathcal{T}_t$ over the current center set $C_t$.

**Main$(Q)$:**

1. Initialize the root $r$ to be empty.

2. For each update $q_t$ at time $t$:

   - **If** $q_t$ is INSERT$(c)$: call INSERT_CENTER$(c, r)$ where $r$ is the root.
   - **If** $q_t$ is DELETE$(c)$: call DELETE_CENTER$(c, r)$ where $r$ is the root.
   - Output the updated tree $\mathcal{T}_t$.

---

**Procedure** REBUILD$(u)$:

1. Let $C_u$ be the current center set at $u$, set anchor $m^u$ be the coordinate-wise median of centers $C_u$. Initialize the main part $u_0 = u$.

2. Initialize an update counter at $u$ to be $\mathrm{Cnt}_u = 0$ and set $k_u = |C_u|$.

3. Compute a sequence of candidate cuts $\{(\omega_t, \rho_t)\}$ using an exponential clock: For each $t$, sample $i_t \in [d]$, $\sigma_t \in \{-1, 1\}$, $Z_t \sim \mathrm{Unif}[0, 2^p]$. Define the cut $\omega_t = (i_t, \vartheta_t)$, where $\vartheta_t = m_{i_t} + \sigma_t(Z_t)^{1/p}$. Assign the timestamps $\rho_t$ as the arrival times of a Poisson process.

4. Iterate over the cuts $\omega_t$ in increasing order of their timestamps $\rho_t$. Accept it iff $\omega_t$ separates two centers in the main part $u_0$. After each accepted cut, update the main part $u_0$ to the side containing the anchor. Stop when the main part contains fewer than $|C_u|/2$ centers. Then, set the stopping time $\rho^u$ to be the timestamp of the last accepted cut,
$$\rho^u = \max\{\rho_t : \text{cut } \omega_t \text{ is accepted}\}.$$

5. Call REBUILD$(v)$ for each leaf $v$ containing more than one center in the subtree rooted at $u$.

---

**Procedure** INSERT_CENTER$(c, u)$:

1. Increment update counter at $u$; if updates exceed $k_u/4$, call REBUILD$(u)$.

2. Get the earliest cut $(\omega, \rho) = $ GET_EARLIEST_CUT$(c)$ that separates $c$ and $m^u$.

3. Let $(\omega_1', \rho_1'), \dots, (\omega_r', \rho_r')$ be cuts used by $u$ and $\rho^u$ be the stopping time.

4. **If** $\rho = \rho_j'$ for some $j$:
   Assign $c$ to node $v$ separated by the cut $\omega_j'$, and call INSERT_CENTER$(c, v)$.

5. **If** $\rho > \rho^u$: Assign $c$ to the main part $u_0$, and call INSERT_CENTER$(c, u_0)$.

6. **If** $\rho \leq \rho^u$ and $\rho \neq \rho_j'$ for every $j$:
   Insert new cut $\omega$ into the sequence of cuts used by $u$, maintaining increasing order by $\rho$. Create a new leaf node containing $c$, and attach it to the tree at the cut point.

> **Procedure** DELETE_CENTER$(c, u)$:
>
> 1. Increment update counter at $u$; if updates exceed $k_u/4$, call REBUILD$(u)$.
> 2. Locate the leaf node $v$ containing $c$.
> 3. **If** the leaf contains only $c$: Remove both the leaf and its parent cut.
> 4. **Else**: Delete $c$ from the leaf $v$ and call DELETE_CENTER$(c, v)$.

Figure 2: Dynamic algorithm for explainable $k$-medians in $\ell_p$

*Proof of Lemma B.1.* We describe an implementation of the static algorithm on the set of centers $C_t$, using specific oracles GET_ANCHOR and STOPPING_ORACLE.

To couple with the dynamic algorithm, we mirror each partition leaf call currently maintained in the dynamic algorithm solution. We begin with the partition leaf call at the root node. Let $t' \le t$ denote the time of the most recent rebuild of this root partition leaf as of time $t$, and let $k_{t'} = |C_{t'}|$ be the number centers present at that rebuild time. Assume both the dynamic and static algorithms use the same infinite sequence of candidate cuts with associated timestamps for the root PARTITION_LEAF call.

For any fixed sequence of cuts with timestamps, let $m^r$ be the anchor and $\rho^r$ be the stopping time used by the dynamic algorithm for this root partition leaf. In the static algorithm, we adopt the same oracles as the dynamic one: the oracle GET_ANCHOR returns $m^r$ and STOPPING_ORACLE returns True if and only if the timestamp of the input cut exceeds $\rho^r$. As a result, the static algorithm accepts exactly the same sequence of cuts as the dynamic algorithm. Therefore, the partial tree rooted at $r$ produced by this PARTITION_LEAF call in the static algorithm is identical to that maintained by the dynamic algorithm. We will show that these two oracles are valid for the static algorithm, which means they satisfy the required properties in Section 2.

We first show that GET_ANCHOR returns an approximate median of centers $C_t$. Because this is the most recent rebuild of the root node $r$, there have been fewer than $k_{t'}/4$ updates since then. Note that the anchor $m^r$ is chosen as the coordinate-wise median of all centers in $C_{t'}$ at time $t'$. For each coordinate $i$, at most half of the centers in $C_{t'}$ lie on either side of $m^r$. Hence, even after $k_{t'}/4$ updates, there remain at most $3k_t/4$ centers in $C_t$ on either side of $m^r$ along every coordinate.[3] Therefore, the anchor $m^r$ remains an approximate median for the current set of centers $C_t$.

We next show that the STOPPING_ORACLE guarantees that when partitioning stops, every leaf contains at most a $3/4$ fraction of centers in $C_t$. Consider any leaf that is separated from the main part during the partitioning. Each such leaf contains only centers that lie on one side of the anchor $m^r$ along the coordinate used by the cut that separates it. Since the anchor $m^r$ is an approximate median of centers in $C_t$, at most $3k/4$ centers lie on either side of $m^r$ along every coordinate. Therefore, each separated leaf contains at most a $3/4$ fraction of centers in $C_t$. As for the main part, recall that at the stopping time $\rho^r$ during the last rebuild, it contains at most $k_{t'}/2$ centers in $C_{t'}$. After at most $k_{t'}/4$ updates, the main part contains at most a $3/4$ fraction of centers in $C_t$.

At each recursive step, we use the same sequence of cuts and adopt the corresponding anchor and stopping time used by the dynamic algorithm. This guarantees that the static algorithm mirrors the behavior of the dynamic one at every level of the recursion. Therefore, the static algorithm constructs exactly the same threshold tree as the dynamic algorithm. This completes the coupling argument and establishes that the output of the dynamic algorithm is identically distributed to that of the static algorithm on input $C_t$.

□

---

[3] Consider any fixed coordinate $i$ and one side of $m_i^r$. The fraction of centers lying on this side of $m^r$ is maximized when all $k_{t'}/4$ updates remove centers from the opposite side. Thus, the fraction of centers lying on this side is at most $2/3$ after updates.

## B.2 Efficient implementation and analysis

In this section, we present a practical implementation of dynamic algorithm as shown in Figure 2. We evaluate the efficiency of the algorithm from two perspectives: update time and recourse.

First, the update time at request $q_t$ refers to the time required to modify the threshold tree $\mathcal{T}_{t-1}$ in response to the $t$-th request $q_t$ (either an insertion or deletion of a center), resulting in a new tree $\mathcal{T}_t$. Second, the recourse at request $q_t$ is defined to be the number of nodes that differ between $\mathcal{T}_{t-1}$ and $\mathcal{T}_t$, i.e., the size of their symmetric difference between the two trees.

We focus on bounding these quantities in the amortized sense, i.e., the total update time and total recourse over all requests, averaged across the requests. The following lemma summarizes the performance guarantees of the dynamic algorithm.

**Lemma B.3.** *Given a sequence of requests, where each request is either an insertion or a deletion of a single center, the dynamic algorithm satisfies with probability $1$ the following guarantees for every $t \geq 1$*

> *1. the amortized recourse is $O(\log k)$,*
>
> *2. the amortized update time is $O(d \log^3 k)$,*

*where $k = \max_{i=1}^{t} |C_i|$.*

We first describe an efficient implementation of the dynamic algorithm. For each node $u$ where REBUILD is called, we maintain a self-balancing binary search tree that stores all cuts with timestamps $(\omega_1', \rho_1'), (\omega_2', \rho_2'), \ldots (\omega_r', \rho_r')$ used in the partial tree rooted at $u$. This data structure enables efficient updates. When a new request arrives to insert or delete a center $c$, we call GET_EARLIEST_CUT$(c)$ to compute the earliest cut that separates $c$ from the anchor $m^u$, and then search the binary search tree to locate where this separation occurs in the partition leaf path of $u$.

We now describe an efficient implementation of the function GET_EARLIEST_CUT. Without loss of generality, we assume that all centers are in $[-1, 1]^d$. The function GET_EARLIEST_CUT takes a center $c$ as input and outputs the earliest cut $(\omega, \rho)$ that separates $c$ from the anchor $m^u$ among a sequence of candidate cuts $(\omega_1, \rho_1), (\omega_2, \rho_2), \ldots$. Each cut $\omega_t = (i_t, \vartheta_t)$ is generated by sampling a coordinate $i$, a sign $\sigma \in \{-1, 1\}$ uniformly at random, and a parameter $\theta \in [0, 2]$ drawn from the distribution with density $f(x) = px^{p-1}/2^p$. The threshold is then set as $\vartheta_t = m_i^u + \sigma\theta$. The associated timestamps $\rho_t$ follow the arrival times of a Poisson Process with rate $\lambda = 1$.

To facilitate efficient implementation, we first observe that the problem naturally decomposes across coordinates. Specifically, for each coordinate $i \in \{1, 2, \cdots, d\}$, we can independently maintain and query the earliest cut that separates $c$ from $m^u$ along coordinate $i$. We then return the cut with the minimum timestamp across all coordinates.

To achieve this, we maintain an independent stream of candidate cuts for each pair of coordinate $i$ and direction $\sigma \in \{-1, 1\}$. Each such stream consists of cuts $\omega = (i, \vartheta)$ where $\vartheta = m_i^u + \sigma\theta$ and the timestamps given by the arrival times of a Poisson process with rate $1/2d$. This decomposition is formally justified by the Coloring Theorem (see, e.g. Kingman (1992), page 53 or Mitzenmacher and Upfal (2017), page 223), which states:

**Theorem B.4** (Coloring Theorem). *Let $\Pi_t$ be a Poisson process on the real line with rate $\lambda$. Assign to each event of the process a color from a finite set $\{1, \cdots, M\}$, where each event is independently colored with probability $p_i$ of receiving color $i$. Then the counts of events of each color, $\Pi_1, \cdots, \Pi_M$, form independent Poisson processes, with rates $\lambda p_1, \cdots, \lambda p_M$, respectively.*

The original sequence of candidate cuts has timestamps given by the arrival times of a Poisson process with rate $1$. Each cut is independently assigned a pair $(i, \sigma)$ with uniform probability $1/2d$ over all $2d$ possible combinations. By the Coloring Theorem, the subset of cuts corresponding to any fixed pair $(i, \sigma)$ forms an independent Poisson process with rate $1/2d$ and these $2d$ streams are independent. Therefore, the union of all these subsequences of cuts has the same distribution as the original sequence of candidate cuts.

We then formulate the earliest cut along each coordinate as the following general problem. We are given a fixed anchor value $m \in [-1, 1]$, and a sequence of random cuts specified by thresholds $\vartheta_t$ drawn from $[m, m+2]$ according to a probability density function $f(x)$, with associated timestamps

$\rho_t$, corresponding to the arrival times of a Poisson process with rate $\lambda_0$. For a query point $y \in [m, 1]$, we aim to find the earliest cut that separates $y$ and $m$, i.e., the cut with the smallest timestamp such that its threshold $\vartheta_t$ lies in $(m, y]$. This formulation arises naturally in our setting, where $\lambda_0 = 1/2d$, the density function $f(x) = p(x - m)^{p-1}/2^p$, $m$ represents the $i$-th coordinate of the anchor, and $y$ corresponds to the $i$-th coordinate of some center $c$. A simple approach for solving this problem is to simulate the sequence of cuts with timestamps and return the first one that lies in $(m, y]$. We refer to this as the static algorithm.

We now describe a data structure that efficiently retrieves the earliest cut along a given coordinate. This data structure maintains a self-balancing binary search tree. Given an anchor $m$ and a set of $k$ values $m \leq y_1 < y_2 < \cdots < y_k \leq 1$, this binary search tree maintains these values in increasing order. Each node in the binary search tree stores a value $y$ along with the earliest cut that separates $y$ from the anchor $m$, including the timestamp of that cut. If the queried value $y$ is present in the tree, the associated earliest separating cut can be retrieved in $O(\log k)$ time.

Now suppose we need to insert a new value $y \in [m, 1]$ into this data structure. Assume the binary search tree currently stores $k$ values $m \leq y_1 < y_2 < \cdots < y_k \leq 1$. We first locate the position of $y$ in the tree in $O(\log k)$ time, either identifying the smallest index $j$ such that $y < y_j$, or determining that $y > y_k$. Let $y_0 = m$. If there exists some $1 \leq j \leq k$ such that $y_{j-1} < y < y_j$, then we first retrieve the earliest cut $(\vartheta, \rho)$ that separates $y_j$ from $m$. We consider two different cases:

1. $y_{j-1} < y < y_j$ for some $1 \leq j \leq k$ and $(\vartheta, \rho)$ also separates $y$ from $m$, (i.e. $\vartheta \leq y$);
2. either $y_{j-1} < y < y_j$ and $(\vartheta, \rho)$ does not separates $y$ from $m$ (i.e. $\vartheta > y$) or $y_k < y \leq 1$.

For the first case, we store this cut $(\vartheta, \rho)$ at the node $y$ as the earliest cut that separates $y$ from $m$.

For the second case, we first sample a new cut as follows. If $y \geq y_k$, then let $y_{j-1} = y_k$. Sample a new threshold $\vartheta' \in (y_{j-1}, y]$ using the weighted density function

$$\tilde{f}(x) = \frac{f(x)}{\Pr\{\vartheta \in (y_{j-1}, y]\}} = \frac{f(x)}{\int_{y_{j-1}}^{y} f(t)\mathrm{d}t}, \ x \in (y_{j-1}, y].$$

We then sample a timestamp for this cut as $\rho' = \rho + z$, if $y \leq y_k$, otherwise if $y > y_k$, $\rho' = z$, where $z \sim \exp(\lambda)$ with rate

$$\lambda = \lambda_0 \cdot \Pr\{\vartheta \in (y_{j-1}, y]\} = \lambda_0 \cdot \int_{y_{j-1}}^{y} f(t)\mathrm{d}t,$$

where $\lambda_0$ is a parameter of the data structure. Let $(\vartheta'', \rho'')$ be the earliest cut that separates $y_{j-1}$ from $m$. We then compare the two cuts and store at node $y$ the one with the smaller timestamp. If $\rho' < \rho''$, then we store the new cut $(\vartheta', \rho')$ at node $y$ as the earliest cut; otherwise, we store the cut $(\vartheta'', \rho'')$.

**Lemma B.5.** *Given a sequence of query points $y_1, y_2, \cdots$, the earliest cuts maintained by the data structure are distributed identically to those returned by the static algorithm.*

*Proof.* We prove this lemma by induction. For the first query point, the data structure and the static algorithm samples the earliest cut that separates this point from the same distribution. We now assume that for the first $k$ query points $y_1, \cdots, y_k$, the earliest cuts returned by the data structure are distributed identically to those returned by the static algorithm. By coupling these two algorithms, we further assume that the data structure and the static algorithm return exactly the same earliest cuts for these query points.

We now consider a new query point $y_{k+1}$ and argue that the earliest cuts returned by two algorithms are distributed identically. Let $y_{(1)}, y_{(2)}, \cdots, y_{(k)}$ be the first $k$ query points sorted in increasing order. Let $y_{(0)} = m$. Suppose this new query point is in the first case, which means there exists $1 \leq j \leq k$ such that $y_{(j-1)} < y_{k+1} < y_{(j)}$ and the earliest cut $(\vartheta_t, \rho_t)$ that separates $y_{(j)}$ maintained by the data structure also separates $y_{k+1}$. Since $y_{k+1} < y_{(j)}$ and this cut $(\vartheta_t, \rho_t)$ is the earliest cut that separates $y_{(j)}$ in the static algorithm, this cut is also the earliest cut for $y_{k+1}$ returned by the static algorithm.

We now consider this new query point is in the second case, either $y_{(j-1)} < y_{k+1} < y_{(j)}$ and the earliest cut $(\vartheta_t, \rho_t)$ that separates $y_{(j)}$ does not separates $y_{k+1}$ or $y_{(k)} < y_{k+1} \leq 1$. If $y_{k+1} > y_{(k)}$, we set $y_{(j-1)} = y_{(k)}$. We decompose the sequence of cuts used in the static algorithm into three

disjoint subsequences. These three subsequences contain all cuts in three disjoint intervals $(m, y_{(j-1)}]$, $(y_{(j-1)}, y_{k+1}]$, and $(y_{k+1}, m+2]$ respectively. By the Coloring Theorem, the timestamps of these subsequences follow the arrival times of three independent Poisson processes. Since the cut is sampled from $(y_{(j-1)}, y_{k+1}]$ with probability $p = \int_{y_{(j-1)}}^{y_{k+1}} f(t)\mathrm{d}t$, the timestamps of all cuts in $(y_{(j-1)}, y_{k+1}]$ follows the arrival times of a Poisson process with rate

$$\lambda = \lambda_0 \cdot \Pr\{\vartheta \in (y_{(j-1)}, y_{k+1}]\} = \lambda_0 \cdot \int_{y_{(j-1)}}^{y_{k+1}} f(t)\mathrm{d}t.$$

Suppose there exists $1 \leq j \leq k$ such that $y_{(j-1)} < y_{k+1} < y_{(j)}$. Since the earliest cut $(\vartheta_t, \rho_t)$ that separates $y_{(j)}$ does not separate $y_{k+1}$ in the static algorithm, the first cut in the interval $(y_{(j-1)}, y_{k+1}]$ must arrive after $\rho_t$. The time of the first arrival of this subsequence follows an exponential distribution with rate $\lambda$. Due to the memoryless property of the exponential distribution, the first arrival of cuts in $(y_{(j-1)}, y_{k+1}]$ follows $\rho_t + z$, where $z \sim \exp(\lambda)$. Suppose $y_{k+1} > y_{(k)}$. Then, the time of the first arrival in this subsequence is $z \sim \exp(\lambda)$. Therefore, in the static algorithm, the first cut in $(y_{(j-1)}, y_{k+1}]$ has the exact same distribution as the new cut sampled in the data structure. If $y_{(j-1)} \neq m$, then the first cut in $(m, y_{(j-1)}]$ is the same in the data structure and the static algorithm. Combining two parts, the earliest cut that separates $y_{k+1}$ returned by the data structure has the same distribution as that returned by the static algorithm. $\square$

*Remark.* The assumption that all centers lie in $[-1, 1]^d$ is made for the ease of exposition. The algorithm can be implemented without this assumption. Under the $\ell_p$ norm, the threshold $\theta$ is drawn from a distribution with density $f(x) = px^{p-1}/R^p$ where $R > y$ is the bounding radius. Conditioned on $x \in (y_{j-1}, y]$, the probability density function becomes

$$\tilde{f}(x) = \frac{f(x)}{\int_{y_{j-1}}^y f(t)\mathrm{d}t} = \frac{p(x-m)^{p-1}}{(y-m)^p - (y_{j-1}-m)^p}.$$

To sample a threshold $\vartheta'$ following this distribution, we draw a uniform random variable $U \in [(y_{j-1}-m)^p, (y-m)^p]$ and set $\vartheta' = U^{1/p}$. Moreover, multiplying all timestamps by the same positive number does not affect the analysis of B.5. Thus, we can equivalently sample $z \sim \exp(\lambda)$ with $\lambda = (y-m)^p - (y_{j-1}-m)^p$, without altering the analysis. With these minor modifications, the algorithm no longer depends on the boundedness assumption that the centers lie in $[-1, 1]^d$.

We now analyze the recourse and the update time of the dynamic algorithm with the above implementation.

*Proof of Lemma B.3.* Fix $t \geq 1$ and condition on the randomness of the algorithm until time $t$. Since the subsequent argument holds for any fixed randomness, the guarantees hold with probability 1.

**Recourse:** Let $\mathcal{R}(i)$ be the recourse incurred by request $i$. We partition the requests into two sets: Let $S_1 \subseteq [t]$ be the set of requests for which the REBUILD operation is not called during the update due to this request. Let $S_2 = [t] \setminus S_1$ be the remaining requests where the REBUILD operation is called. We analyze each case separately.

Case 1 ($i \in S_1$): In this case, the request does not trigger a REBUILD operation, and the recourse is at most $\mathcal{R}(i) \leq 2$. This is because if the request is an insertion, at most two nodes are added to $\mathcal{T}_{i-1}$; if it is a deletion, at most two nodes are removed, i.e., the leaf that contains the center $c$ and its parent in both cases. As a result, the total recourse over all such requests is bounded by

$$\sum_{i \in S_1} \mathcal{R}(i) \leq 2|S_1| \leq 2t. \tag{2}$$

Case 2 ($i \in S_2$): The REBUILD will only be called on one node $u_i$ for each request $i$. Let $C_{u_i}$ be the set of centers contained in the node $u_i$ of $\mathcal{T}_{i-1}$, and let $k' = |C_{u_i}|$. Since REBUILD($u_i$) is called, all $2k' - 1$ nodes in the subtree rooted at $u_i$ are removed from $\mathcal{T}_{i-1}$. If the request $i$ is an insertion of a center $c$, a new threshold tree is constructed at $u_i$ using the updated center set $C_{u_i} \cup \{c\}$, which has size $k' + 1$. This results in inserting $2(k'+1) - 1 = 2k' + 1$ nodes back into the tree. Therefore, the recourse is $\mathcal{R}(i) = 2k' - 1 + 2k' + 1 = 4k'$. If the request $i$ is a deletion of a center $c$, the updated center set is $C_{u_i} \setminus c$ of size $k' - 1$, and the rebuilt threshold tree contains $2(k'-1) - 1 = 2k' - 3$

nodes. The recourse in this case is $\mathcal{R}(i) = (2k' - 1) + (2k' - 3) = 4k' - 4$. In either case, we have the bound $\mathcal{R}(i) \le 4k'$.

We now analyze the total recourse for $S_2$. Each node $u$ on which the algorithm calls a REBUILD stores an update counter $\text{Cnt}_u$. This update counter is initialized to zero when the node is rebuilt and is incremented by one each time an update (insertion or deletion) involves node $u$. This node $u$ also stores the number of centers $k_u$ in this node when it is rebuilt. Since the dynamic algorithm rebuilds this node $u$ after $k_u/4$ updates, we have $k' \le k_u + k_u/4$. Therefore, we have $\text{Cnt}_{u_i} = k_{u_i}/4 \ge k'/5$. Hence, we have

$$\sum_{i \in S_2} \mathcal{R}(i) \le \sum_{i \in S_2} 20 \cdot \text{Cnt}(u_i). \tag{3}$$

The right-hand side of (3) is bounded by the total number of times any node's counter is incremented. According to the analysis in Lemma B.1, the dynamic algorithm guarantees that after the partition leaf call of a node $u$, each leaf has at most a $3/4$ fraction of the centers contained in $u$. Let $k = \max_{i=1}^t |C_i|$ be the maximum number of centers during the first $t$ requests. Therefore, each update request is involved in at most $O(\log k)$ calls of INSERT_CENTER or DELETE_CENTER. Thus, the total number of times any node's counter is incremented is bounded by $O(t \log k)$. Combining this with (2) and (3), we conclude that $\sum_{i=1}^t \mathcal{R}(i) = O(t \log k)$ and thus the amortized recourse is $O(\log k)$.

**Update Time:** As in the amortized recourse analysis, let $S_1 \subseteq [t]$ be the set of time steps where REBUILD is called on some node $u_i$, and let $S_2 = [t] \setminus S_1$. We now split the analysis into two cases, depending on whether or not a rebuild is triggered.

Case 1 ($i \in S_1$): Suppose the request $i$ is an insertion of center $c_i$. Let $u_1, u_2, \ldots, u_l$ be the nodes for which INSERT_CENTER$(c_i, u_j)$ is called. Each such call on node $u$ takes $O(d \log k)$ time, where $k = \max_{i=1}^t |C_i|$. It takes

- $O(d \log k)$ time to update the $d$ self-balancing binary search trees stored in $u$;

- $O(d \log k)$ time to compute the earliest cut through GET_EARLIEST_CUT$(c_i)$;

- $O(\log k)$ time to locate this earliest cut and insert the center by searching the self-balancing binary search tree that maintains all cuts $(\omega_1', \rho_1'), (\omega_2', \rho_2'), \ldots, (\omega_r', \rho_r')$ currently used in the partition leaf call of $u$.

Since the center $c_i$ is involved in at most $O(\log k)$ INSERT_CENTER calls, the update time for an insertion request $i \in S_1$ is $\text{Time}(i) = O(d \log^2 k)$. The same asymptotic bound holds for deletions, as finding the leaf that contains the deletion center $c_i$ takes $O(d \log^2 k)$ time, and the removal takes constant time. Thus, we have

$$\sum_{i \in S_1} \text{Time}(i) = O(|S_1| \cdot d \log^2 k). \tag{4}$$

Case 2 ($i \in S_2$): Let $u_i$ be the node that is rebuilt at request $i$. As in Case 1, the time to process the request before the rebuild is $O(d \log^2 k)$. If $u_i$ contains $k'$ centers at this request $i$, then REBUILD$(u_i)$ takes $O(k'd \log^2 k)$ time.

Since when REBUILD$(u_i)$ is triggered, we have the update counter $\text{Cnt}_{u_i} \ge k'/5$. Thus, we charge the rebuild time to the update counter. That is the update time $\text{Time}(i) \le O(\text{Cnt}_{u_i} \cdot d \log^2 k)$. Therefore, we have

$$\sum_{i \in S_2} \text{Time}(i) \le O(d \log^2 k) \cdot \sum_{i \in S_2} \text{Cnt}_{u_i}. \tag{5}$$

By the analysis in recourse, we have $\sum_{i \in S_2} \text{Cnt}_{u_t} \le O(t \log k)$. Combining (4) and (5), we obtain that the total update time is at most

$$\sum_{i=1}^t \text{Time}(i) = O(td \log^3 k)$$

and so the amortized update time is $O(d \log^3 k)$. $\qquad\square$

We now prove the main theorem of the dynamic algorithm.

*Proof of Theorem 5.1.* By Corollary B.2 and Lemma B.3, we get the approximation guarantee, amortized recourse, and the amortized update time of the dynamic algorithm. □

## B.3 Fully Dynamic Explainable Clustering Algorithm

In this section, we provide a fully dynamic explainable clustering algorithm for the setting in which the clustering data set evolves over time through insertions or deletions of data points. This algorithm maintains an explainable $k$-clustering that is competitive against the optimal (unconstrained) $k$-clustering. This setting contrasts with Sections B.1 and B.2, where the cluster centers change over time.

Formally, the input is a stream of updates on the data set, where each update is an insertion or deletion of a data point. This generates a sequence of datasets $X_1, X_2, \ldots$. If $t$ is an insertion request of a new data point $x_t$, then $X_t = X_{t-1} \cup \{x_t\}$, whereas if $t$ is a deletion request of an existing data point $x_t \in X_{t-1}$, then $X_t = X_{t-1} \setminus \{x_t\}$. We obtain our fully dynamic explainable clustering algorithm by combining our dynamic algorithm from Section 5 with the fully dynamic $k$-medians algorithm of Bhattacharya et al. (2025). This fully dynamic $k$-medians algorithm maintains a constant-factor approximation while changing only $\tilde{O}(1)$ centers per update.

**Corollary B.6.** *Given a positive integer $k$ and a stream of updates that are insertion or deletion requests of data points in $\mathbb{R}^d$, for every $p \geq 1$ there exists a fully-dynamic explainable clustering algorithm that outputs a threshold tree $\mathcal{T}_t$ for every $t \geq 1$ satisfying*

1. $\mathbf{E}[\mathrm{cost}_p(X_t, \mathcal{T}_t)] \leq O\left(p(\log k)^{1 + \frac{1}{p} - \frac{1}{p^2}} \log \log k\right) \mathrm{OPT}_{k,p}(X_t)$,

2. *the expected amortized update time is $\tilde{O}(kd + (\log \Delta)^2 d \log^3 k)$,*

3. *the expected amortized recourse is $O((\log \Delta)^2 \log k)$*

*where $\Delta$ is the aspect ratio[4] of all data points in $X = \bigcup_{i=1}^t X_i$, $\mathrm{OPT}_{k,p}(X_t)$ is the $\ell_p$ cost of an optimal (unconstrained) $k$-medians clustering of $X_t$ and $\tilde{O}$ hides polylogarithmic factors in $\Delta, k$ and $n = |X|$.*

To prove Corollary B.6, we first show how to combine any fully-dynamic (unconstrained) $k$-medians clustering algorithm under the $\ell_p$ norm with our dynamic algorithm from Section 5 to get a fully-dynamic explainable clustering algorithm.

**Definition B.7.** *An algorithm $\mathcal{A}$ is an $(\alpha, u, r)$ dynamic $k$-medians clustering algorithm under the $\ell_p$ norm, if for every stream of updates that are insertion or deletion requests of data points, the algorithm outputs $k$ centers $C_t$ after each update $t$, such that $\mathbf{E}[\mathrm{cost}_p(X_t, \mathcal{T}_t)] \leq \alpha \, \mathrm{OPT}_{k,p}(X_t)$, the expected amortized update time is $u$ and the expected amortized recourse is $r$.*

Fix an iteration $t$ of an $(\alpha, u, r)$ dynamic $k$-medians clustering algorithm under the $\ell_p$ norm for $p \geq 1$. After processing the $t$-th update request, the algorithm updates the current set of centers from $C_{t-1}$ to $C_t$. To apply Theorem 5.1, we treat each $c \in C_{t-1} \setminus C_t$ as a deletion from the current center set $C_{t-1}$ and each $c \in C_t \setminus C_{t-1}$ as an insertion into it. Algorithm 3 formalizes this procedure, and its performance guarantees are proved in Proposition B.8.

**Proposition B.8.** *Given a positive integer $k$, a stream of updates that are insertion or deletion requests of data points in $\mathbb{R}^d$, and an $(\alpha, u, r)$ dynamic $k$-medians clustering algorithm $\mathcal{A}$ under the $\ell_p$ norm for some $p \geq 1$, Algorithm 3 outputs a threshold tree $\mathcal{T}_t$ for every time $t \geq 1$ satisfying*

1. $\mathbf{E}[\mathrm{cost}_p(X_t, \mathcal{T}_t)] \leq O\left(\alpha \cdot p(\log k)^{1 + \frac{1}{p} - \frac{1}{p^2}} \log \log k\right) \mathrm{OPT}_{k,p}(X_t)$

2. *the expected amortized update time is $O(u + r \cdot d \log^3 k)$*

3. *the expected amortized recourse is $O(r \cdot \log k)$.*

---

[4]The aspect ratio of a set of points $X$ under $\ell_p$ norm is $\Delta = \frac{\max_{x,y \in X} \|x - y\|_p}{\min_{x,y \in X, x \neq y} \|x - y\|_p}$.



**Algorithm** FULLY_DYNAMIC_PARTITION_LEAF

**Input**: an integer $k$, a number $p \geq 1$, a stream of update requests of data points $q_1, q_2, \ldots$ and an $(\alpha, u, r)$ dynamic $k$-medians clustering algorithm $\mathcal{A}$ under the $\ell_p$ norm.

**Output**: threshold trees $\mathcal{T}_1, \mathcal{T}_2, \ldots$

1. Initialize the root $root$ to be empty.

2. Initialize $C_0$ to be an empty set of centers.

3. For every $t \geq 1$:

   - Run algorithm $\mathcal{A}$ to process request $q_t$ and get a new set of centers $C_t$.
   - For every center $c \in C_{t-1} \setminus C_t$:
     Call DELETE_CENTER$(c, root)$ in Figure 2.
   - For every center $c \in C_t \setminus C_{t-1}$:
     Call INSERT_CENTER$(c, root)$ in Figure 2.
   - Output the threshold tree $\mathcal{T}_t$ rooted at $root$.



Figure 3: Fully Dynamic algorithm for explainable $k$-medians in $\ell_p$

Before we prove Proposition B.8, we show how it yields Corollary B.6 by choosing the fully dynamic $k$-medians algorithm $\mathcal{A}$ by Bhattacharya et al. (2025).

*Proof of Corollary B.6.* The dynamic algorithm for $k$-medians from Bhattacharya et al. (2025) achieves an $O(1)$ approximation. It has $O(\log^2 \Delta)$ expected amortized recourse and $\tilde{O}(kd)$ expected amortized update time.[5] As a result, by Proposition B.8, we get the conclusion. $\square$

We proceed to prove Proposition B.8.

*Proof of Proposition B.8.* Fix any $t \geq 1$. For every $i \in \{1, 2, \ldots, t\}$, let $C_i$ denote the set of centers produced by $\mathcal{A}$ after processing the $i$-th request. Let $r_i = |C_i \triangle C_{i-1}|$ denote the recourse at time $i$. During iteration $i$, Algorithm 3 produces $r_i$ intermediate center sets $C'_{i,1}, C'_{i,2}, \ldots, C'_{i,r_i} = C_i$ corresponding to the individual center update requests applied to $C_{i-1}$. Since deletions are processed before insertions, each intermediate set has size at most $k$. Let $\mathcal{T}'_{i,1}, \mathcal{T}'_{i,2}, \ldots, \mathcal{T}'_{i,r_i}$ denote the intermediate threshold trees produced by Algorithm 3 after each center update during iteration $i$. For the rest of the proof, we condition on a fixed sequence of center sets $C'_{1,1}, C'_{1,2}, \ldots, C'_{t,r_t} = C_t$.

**Approximation:** Applying Theorem 5.1, for every $i \in \{1, 2, \ldots t\}$ and $j \in \{1, 2, \ldots, r_i\}$ the following inequality holds:

$$\mathbf{E}[\text{cost}_p(X_i, \mathcal{T}'_{i,j}) \mid C'_{1,1}, \ldots, C'_{t,r_t}] \leq O\left(p(\log k)^{1 + \frac{1}{p} - \frac{1}{p^2}} \log \log k\right) \text{cost}_p(X_i, C'_{i,j}).$$

Therefore, choosing $j = r_t$ we obtain

$$\mathbf{E}[\text{cost}_p(X_t, \mathcal{T}_t) \mid C'_{1,1}, \ldots, C'_{t,r_t}] \leq O\left(p(\log k)^{1 + \frac{1}{p} - \frac{1}{p^2}} \log \log k\right) \text{cost}_p(X_t, C_t).$$

Taking the expectation at both sides of the inequality and using the fact that $\mathcal{A}$ is an $\alpha$-approximation algorithm, the approximation guarantee follows.

**Recourse**: By Theorem 5.1, the amortized recourse of DYNAMIC_PARTITION_LEAF is $O(\log k)$ with probability 1. Hence, after processing $t$ requests, the total number of tree nodes modified is $O(R \log k)$, where $R = \sum_{i=1}^{t} r_i$ denotes the total recourse of algorithm $\mathcal{A}$, i.e., the total number of center update requests. Therefore, the expected total number of tree nodes modified up to the $t$-th request is $O(\mathbf{E}[R] \log k) = O(rt \log k)$, which corresponds to the expected total recourse. Dividing by $t$, we obtain the expected amortized recourse of $O(r \log k)$.

---

[5]The algorithm introduced in Bhattacharya et al. (2025) is aimed for the metric $k$-medians problem and it's amortized update time is $\tilde{O}(k)$. For our purposes, the amortized update time incurs an extra $O(d)$ factor to calculate the $\ell_p$ distances. $\tilde{O}$ hides polylogarithmic factors in $n, \Delta$, and $k$.

**Update Time**: The total update time of Algorithm 3 equals the sum of the running time of $\mathcal{A}$ for processing all requests and the time taken by DYNAMIC_PARTITION_LEAF to handle all $R = \sum_{i=1}^{t} r_i$ center update requests. By Theorem 5.1, the amortized update time of DYNAMIC_PARTITION_LEAF is $O(d \log^3 k)$ with probability 1. Thus, the total update time is $O(U + Rd \log^3 k)$, where $U = \sum_{i=1}^{t} u_t$ is the total running time of $\mathcal{A}$. Since the expected amortized update time and recourse of $\mathcal{A}$ are $u$ and $r$ respectively, the total expected update time of Algorithm 3 is $O(ut + rt \cdot d \log^3 k)$ and the expected amortized update time guarantee follows. $\qquad \square$

## C  Lower bound for universal algorithms

In this section, we provide a lower bound on the competitive ratio for any universal explainable clustering algorithm. A universal algorithm is required to output a distribution over threshold trees that perform well for all $p \geq 1$ without the prior knowledge of $p$.

Our algorithm for explainable $k$-medians clustering under $\ell_p$ norm samples threshold cuts from a carefully designed distribution that depends crucially on $p$. A natural question is whether there exists an explainable clustering algorithm that is independent of $p$ while achieving a good approximation to the optimal $\ell_p$ cost for all $p \geq 1$ simultaneously. We answer this question in the negative by showing an $\Omega(d^{1/4})$ lower bound on the worst-case competitive ratio of any universal explainable clustering algorithm.

**Theorem 4.2.** *There exists an instance $X \subseteq \mathbb{R}^d$, such that for any distribution over threshold trees, the expected competitive ratio is at least $\Omega(d^{1/4})$ for some $p \geq 1$.*

*Proof.* The instance has two centers, one at the origin $c_1 = (0, 0, \ldots, 0)$, and the other at $c_2 = (1 + d^{3/4}, 1, \ldots, 1)$, along with many data points co-located at each center and one special point $x = (1, 1, \ldots, 1)$. We show that any distribution $D$ over threshold trees (a single threshold cut in this case) yields an explainable clustering such that either the $\ell_1$ or the $\ell_2$ cost is in expectation $\Omega(d^{1/4})$ times the corresponding unconstrained clustering cost.

Case 1: If distribution $D$ assigns $x$ to $c_1$ with probability at least $1/2$, then the expected $\ell_1$ cost of the explainable clustering is at least $d/2$, while the optimal $\ell_1$ clustering cost is $d^{3/4}$ (by assigning $x$ to $c_2$).

Case 2: If distribution $D$ assigns $x$ to $c_2$ with probability at least $1/2$, the expected $\ell_2$ cost of the explainable clustering is at least $d^{3/4}/2$, while the optimal $\ell_2$ clustering cost is $\sqrt{d}$ (by assigning $x$ to $c_1$). $\qquad \square$

## D  Lower bound for explainable $k$-medians under $\ell_p$ norm

In this section, we present a lower bound on the competitive ratio for the explainable $k$-medians problem under $\ell_p$ norm for all $p \geq 1$. In particular, we extend the lower bound instance for explainable $k$-medians clustering under $\ell_2$ norm in Makarychev and Shan (2021) to $\ell_p$ norm for all $p \geq 1$.

**Theorem 4.1.** *For every $p \geq 1$, there exists an instance $X \subseteq \mathbb{R}^d$, such that for every threshold tree $\mathcal{T}$, its clustering cost is at least $\mathrm{cost}_p(X, \mathcal{T}) = \Omega(\log k)\mathrm{OPT}_{k,p}(X)$, where $\mathrm{OPT}_{k,p}(X)$ is the $\ell_p$ cost of the optimal (unconstrained) $k$-medians clustering of $X$.*

We construct the lower bound instance $X$ as follows. Consider the grid $\mathcal{G} = \{0, \epsilon, 2\epsilon, \ldots, 1\}^d$ that is obtained by discretizing the hypercube, where $d = \lceil 64p^4 \ln k \rceil$ and $\epsilon = 1/\ln k$. We choose $k$ centers $C$ uniformly at random from the grid $\mathcal{G}$ and for each $c \in C$, we place two data points $x_{c1} = c + (\epsilon, \epsilon, \ldots, \epsilon)$ and $x_{c2} = c - (\epsilon, \epsilon, \ldots, \epsilon)$. Moreover, for every $c \in C$, we place $n$ data points $x_{cj}, j = 3, 4, \ldots, n + 2$ that coincide with $c$ (i.e. $x_{cj} = c$). We will show that the clustering instance $X = \bigcup_{c \in C} \{x_{cj}, j \in [n + 2]\}$ satisfies with positive probability two properties captured by Lemma D.1 and Lemma D.2 and then show that these properties suffice to prove Theorem 4.1.

The first property we show is that the with high probability all centers in the random set $C$ are well separated.

**Lemma D.1.** *With probability at least $1 - \frac{1}{k^2}$, for any two distinct centers $c, c' \in C$, it holds that* $\|c - c'\|_p \geq \frac{d^{\frac{1}{p}}}{12}$.

*Proof of Lemma D.1.* An equivalent way to choose a center from the grid $\{0, \epsilon, 2\epsilon, \ldots, 1\}^d$ uniformly at random, is to first choose $\tilde{c} \in [-\frac{\epsilon}{2}, 1 + \frac{\epsilon}{2}]^d$ uniformly at random and then choose $c$ to be the closest center of $\tilde{c}$ in the grid. Consider $c, c' \in C$ be two distinct centers of the instance and let $\tilde{c}$ and $\tilde{c}'$ be their corresponding uniform random variables in $[-\frac{\epsilon}{2}, 1 + \frac{\epsilon}{2}]^d$. We have

$$\mathbf{E}[\|\tilde{c} - \tilde{c}'\|_p^p] = \sum_{i=1}^d \mathbf{E}[|\tilde{c}_i - \tilde{c}_i'|^p] = \frac{2d(1+\epsilon)^p}{(p+1)(p+2)},$$

where we used that for each coordinate $i$, $\tilde{c}_i$ and $\tilde{c}_i'$ are independent uniform random variables in $[-\frac{\epsilon}{2}, 1 + \frac{\epsilon}{2}]$. Moreover, the variables $|\tilde{c}_i - \tilde{c}_i'|^p$ are independent for different $i$ and are bounded in $[0, (1+\epsilon)^p]$. By Hoeffding's inequality, we have

$$\Pr\left\{\|\tilde{c} - \tilde{c}'\|_p^p \leq \frac{2d(1+\epsilon)^p}{(p+1)(p+2)} - (1+\epsilon)^p\sqrt{2d \ln k}\right\} \leq \frac{1}{k^4}.$$

Because $d \geq 64\, p^4 \ln k$, we get that $(1+\epsilon)^p\sqrt{2d \ln k} \leq \frac{d(1+\epsilon)^p}{(p+1)(p+2)}$, thus

$$\Pr\left\{\|\tilde{c} - \tilde{c}'\|_p^p \leq \frac{d(1+\epsilon)^p}{(p+1)(p+2)}\right\} \leq \frac{1}{k^4}. \tag{6}$$

This means that with probability at least $1 - 1/k^4$,

$$\|\tilde{c} - \tilde{c}'\|_p \geq \frac{(1+\epsilon)d^{\frac{1}{p}}}{(p+1)^{\frac{1}{p}}(p+2)^{\frac{1}{p}}}.$$

Because $c$ is the closest point in the grid $\mathcal{G}$ to $\tilde{c}$, then $\|c - \tilde{c}\|_p \leq \frac{\epsilon}{2}d^{\frac{1}{p}}$ (the same holds for $c'$ and $\tilde{c}'$). Thus, by the triangle inequality

$$\|c - c'\|_p \geq \frac{(1+\epsilon)d^{\frac{1}{p}}}{(p+1)^{\frac{1}{p}}(p+2)^{\frac{1}{p}}} - \epsilon d^{\frac{1}{p}} \geq \frac{d^{\frac{1}{p}}}{12}.$$

The second inequality holds for sufficiently large $k$, since $\epsilon = 1/\ln k$ can be made arbitrarily small by increasing $k$, and because the function $((p+1)(p+2))^{1/p}$ is decreasing for $p \geq 1$ and thus attains its maximum value 6 at $p = 1$. By applying the union bound over all pairs of centers in $C$, the claim follows. $\square$

To describe the second property, we introduce some notation. Consider a threshold tree $\mathcal{T}$ and a node $u$ of this tree. Let $F_u \subseteq C$ be the set of *undamaged* centers contained in $u$, i.e. the set of centers $c$ in the node such that all the points in the optimal cluster of $c$ are contained in the node $u$. We also define a *path sequence* as any sequence of tuples $(i_1, \theta_1, \sigma_1), (i_2, \theta_2, \sigma_2), \ldots (i_t, \theta_t, \sigma_t)$, such that $t \geq 1$ is an integer, $i_j \in [d]$, $\theta_j \in \mathbb{R}$ and $\sigma_j \in \{\pm 1\}$. Note that any node $u$ is fully specified by the path from the root of $\mathcal{T}$ to $u$ and thus by a path sequence $\pi(u)$, where $(i_j, \theta_j)$ is the $j$-th threshold cut in the path and $\sigma_j$ indicates the direction of the next node in the path. Inversely, for a given path sequence $\pi$ we denote $u(\pi)$ as the node that $\pi$ specifies, i.e.

$$u(\pi) = \bigcap_{(i,\theta,\sigma) \in \pi} \{x \in \mathbb{R} : \sigma(x_i - \theta) \geq 0\}.$$

**Lemma D.2.** *With probability at least $1 - \frac{1}{k}$, for every $t \leq \frac{\log_2 k}{4}$, for every path sequence $\pi = (i_1, \theta_1, \sigma_1), \ldots (i_t, \theta_t, \sigma_t)$ with $i_j \in [d], \theta_j \in [0, 1], \sigma_j \in \{\pm 1\}$, one of the following holds:*

1. *the number of undamaged centers in $u(\pi)$ is at most $|F_{u(\pi)}| \leq \sqrt{k}$; or*

2. *any cut that separates two centers in $u(\pi)$ damages at least $\epsilon|F_{u(\pi)}|/2$ centers in $F_{u(\pi)}$.*

*Proof of Lemma D.2.* It suffices to prove the lemma for path sequences such that $\theta_j \in \{\frac{\epsilon}{2}, \frac{3\epsilon}{2}, \ldots, 1-\frac{\epsilon}{2}\}$. This restriction is without loss of generality, since for every coordinate $i \in [d]$ and for every $r \in \{0, 1, \ldots, \frac{1}{\epsilon}\}$, all the cuts in the interval $(r\epsilon, (r+1)\epsilon]$ are equivalent, in the sense that they induce the same partition of the grid points and thus of the instance $X$.

Fix any path sequence $\pi$ of size $t \leq \frac{\log_2 k}{4}$ and denote $u = u(\pi)$ for simplicity. Assume that the total number of undamaged centers in $u$ is $|F_u| = k' > \sqrt{k}$. Given a threshold cut $\omega = (i, \theta)$, we define $Z_\omega$ to be the number of undamaged centers $c \in F_u$ that are damaged by $\omega$. Conditioned on $|F_u| = k'$, the undamaged centers contained in $u$ are distributed as $k'$ points drawn independently and uniformly from the grid points $\mathcal{G}$ inside $u$, excluding the leftmost and rightmost grid points in each coordinate. Consider each undamaged center $c \in F(u)$. The new cut $\omega$ damages this center $c$ if and only if $c_i \in \{\theta - \epsilon/2, \theta + \epsilon/2\}$. Since there are at most $1/\epsilon$ possible grid positions for $c_i$, this undamaged center $c$ is damaged by the cut $\omega$ with probability at least $\epsilon$. Therefore, we have

$$\mathbf{E}[Z_\omega \mid |F_u| = k'] \geq \epsilon k',$$

where the expectation is taken over the randomness of centers in $F(u)$. Thus, by the Chernoff bound

$$\Pr\left\{Z_\omega \leq \frac{\epsilon}{2}k' \;\middle|\; |F_u| = k'\right\} \leq e^{-\frac{\epsilon k'}{8}} \leq e^{-\frac{\epsilon\sqrt{k}}{8}}.$$

By taking the union bound over all possible cuts in $u$ (at most $d/\epsilon = O(p^4 \ln^2 k)$ in total), we obtain some cut damages less than $\epsilon k'/2$ undamaged centers in $F(u)$ with probability at most $e^{-\frac{\epsilon\sqrt{k}}{16}}$ for sufficiently large $k$. Thus, the probability that both $(1)$ and $(2)$ do not hold is at most $e^{-\frac{\epsilon\sqrt{k}}{16}}$. Moreover, the number of different path sequences at a fixed size $t$ is at most $\left(\frac{2d}{\epsilon}\right)^t = O(p^{4t} \ln^{2t} k)$. Thus, by taking the union bound over all possible path sequences for every $t \leq \frac{\log_2 k}{4}$, the probability that both $(a)$ and $(b)$ do not hold is at most

$$\frac{\log_2 k}{4}\left(\frac{2d}{\epsilon}\right)^{\frac{\log_2 k}{4}} e^{-\frac{\epsilon\sqrt{k}}{16}} = e^{O(\log(p^2 \log k) \log k)} e^{-\frac{\epsilon\sqrt{k}}{16}} \leq \frac{1}{k},$$

where the inequality holds for any fixed $p$ when $k$ is sufficiently large. $\qquad\square$

By Lemma D.1 and D.2 there exists an instance $X$ with $k$ centers and $d = \lceil 64p^4 \ln k \rceil$ such that both properties of these lemmas hold. Moreover, the optimal clustering has $\ell_p$ cost $\mathrm{OPT}_{k,p} \leq 2k\epsilon d^{\frac{1}{p}}$, as we can assign each data point $x_{cj}$ to center $c$. Consider any threshold tree $\mathcal{T}$ with $k$ leaves. We will show that $\mathrm{cost}_p(X, \mathcal{T}) = \Omega(\log k)\mathrm{OPT}_{k,p}$.

First, we consider the case where $\mathcal{T}$ does not separate all centers in $C$, that is, there exists a leaf of the tree that contains two centers $c$ and $c'$. Note that there are $n$ data points located at each of the centers $c$ and $c'$. Hence, the cost of this leaf is at least $n\|c - c'\|_p/2 \geq nd^{\frac{1}{p}}/24$ by Lemma D.1. This cost can be arbitrarily large since $n$ can be arbitrarily large.

Next, consider the threshold tree $\mathcal{T}$ in which each leaf contains exactly one center from $C$. We divide it into the following two cases. In the first case, suppose there exists a level $1 \leq t \leq \frac{\log_2 k}{4}$ that contains at least $\frac{k}{2}$ damaged centers. For each damaged center, there is a data point that was assigned to it in the optimal solution but is reassigned to another center by $\mathcal{T}$. Each such reassignment incurs a cost of $\Omega(d^{1/p})$. Thus, the total cost of $\mathcal{T}$ is at least $\Omega(\frac{k}{2}d^{1/p}) = \Omega(\log k)\mathrm{OPT}_{k,p}$ since $\epsilon = 1/\ln k$.

In the second case, assume that for every $1 \leq t \leq \frac{\log_2 k}{4}$, the number of undamaged centers at level $t$ of $\mathcal{T}$ is at most $\frac{k}{2}$. We call a node $u$ *small* if it contains at most $\sqrt{k}$ undamaged centers, and *large* otherwise. Fix any $t$ in $\{1, 2, \ldots \lfloor\frac{\log_2 k}{4}\rfloor\}$. Since the total number of nodes at level $t$ is at most $k^{\frac{1}{4}}$, the small nodes together contain at most $k^{\frac{3}{4}}$ undamaged centers. Hence, the large nodes contain at least $\frac{k}{2} - k^{\frac{3}{4}} \geq \frac{k}{4}$ undamaged centers for sufficiently large $k$. Because $\mathcal{T}$ contains exactly one center from $C$, all thresholds of cuts lie within $[0, 1]$. By Lemma D.2, the number of undamaged centers that become damaged at level $t$ of $\mathcal{T}$ is at least $\frac{\epsilon k}{4}$. Since each damaged center incurs a reassignment cost of $\Omega(d^{1/p})$ by Lemma D.1, the total cost at level $t$ is $\Omega(\epsilon k d^{1/p})$. By summing over all levels $1 \leq t \leq \frac{\log_2 k}{4}$, the total cost is

$$\Omega\left(\log k \cdot \epsilon k d^{\frac{1}{p}}\right) = \Omega(\log k)\mathrm{OPT}_{k,p}.$$

