# OpenReview forum: "Dynamic Algorithm for Explainable $k$-medians Clustering under $\ell_p$ Norm"
_NeurIPS.cc/2025/Conference — NeurIPS 2025 spotlight_

### Official Review · Reviewer_tdu1 · 2025-06-28

**Clarity:** 3
**Significance:** 3
**Originality:** 3
**Rating:** 5
**Confidence:** 3

**Summary:**

This paper studies the problem of constructing a threshold decision tree that partitions data into k clusters while minimizing the k-median objective under the l_p  norm. Unlike the classical k-median problem, where each node is assigned to the nearest center based on distance, this approach assigns nodes based on decision rules within a tree structure. Each internal node compares a feature value to a threshold and directs the data point to either the left or right subtree, continuing recursively until it reaches a leaf node, which represents a cluster. This tree-based clustering method enhances interpretability by making the clustering structure more transparent and easier to understand.
The authors propose an algorithm with an approximation ratio of O(p⋅log^{1+1/p −1/p^2} k⋅log log k) under the l_p norm for all p≥1, improving upon existing bounds for p=2. Additionally, the paper extends the approach to dynamic clustering, supporting both insertions and deletions of nodes. The dynamic algorithm maintains the same approximation ratio, with an amortized update time (i.e., threshold changes) of O(d log^3 k) and an amortized recourse (i.e., the number of nodes changes to the tree structure) of O(log k).

**Questions:**

I am curious about the explainable clustering problem in the setting where the centers are given as part of the input. Is it possible to construct a single clustering tree that performs well across all l_p norms, rather than building a separate explainable tree for each specific norm? Why or why not?

Additionally, regarding the dynamic clustering algorithm, normalization is applied to all centers. Will normalization also need to be applied to all nodes whenever new nodes are inserted? If so, how does this affect the efficiency of the algorithm?

**Ethical Concerns:**

["NO or VERY MINOR ethics concerns only"]

**Limitations:**

Yes.

**Quality:**

3

**Strengths And Weaknesses:**

Strengths:
The explainable clustering problem is both important and interesting. The authors extend the algorithm to handle the l_p norm and improve upon the best-known approximation ratio for p=2. The techniques and analysis are solid and well-justified. The extension to the dynamic setting is also a valuable contribution—particularly in low-dimensional spaces, where the results are especially strong and practical. I have not read the entire proof in detail, but it appears to be logically sound.

Weaknesses:
The authors do not discuss lower bounds in detail, nor do they attempt to establish any lower bounds for the l_p norms in either the static or dynamic settings.

Small comments:
Line 140: O(p⋅log^{1+1/p −1/p^2} ⋅log log k) -> O(p⋅log^{1+1/p −1/p^2} k⋅log log k), miss k.
Line 191: m_i^u ->  m_{i_t}^u

---

> ### Author Rebuttal · Authors · 2025-07-29
>
> We thank the reviewer for the detailed comments and insightful questions.
>
> W1: We have an example that shows an $\Omega(\log k)$ lower bound on the price of explainability for $k$‑medians under the $\ell_p$ norm for all $p \geq 1$. We will include it in the revised version of the paper. We will also expand the discussion of related lower bounds and hardness of approximation results, including the $\Omega(\log k)$ lower bound on the price of explainability for $k$‑medians under the $\ell_2$ norm due to Makarychev and Shan (2021) and the hardness results for explainable $k$‑medians under the $\ell_1$ norm by Gupta, Pittu, Svensson, and Yuan (2023).
>
> Q1. I am curious about the explainable clustering problem in the setting where the centers are given as part of the input. Is it possible to construct a single clustering tree that performs well across all l_p norms, rather than building a separate explainable tree for each specific norm? Why or why not?
>
> Thank you for the excellent question! There exists an instance with two centers such that no single threshold tree (even distribution over threshold trees) can achieve better than O(d^{1/4}) approximation for all $\ell_p$ norms simultaneously. We will include this example in the revised version.
>
> The instance has two centers, one at the origin $c_1=(0,0,\dots,0)$, and the other at $c_2=(1 + d^{\frac{3}{4}},1,\dots,1)$, along with many data points co-located at each center and one special point $x = (1,1,\dots, 1)$. We show that any distribution $D$ over threshold trees (a single threshold cut in this case) yields an explainable clustering such that either the $\ell_1$ or the $\ell_2$ cost is in expectation $\Omega(d^{\frac{1}{4}})$ times the corresponding unconstrained clustering cost.
>
> Case 1: If distribution $D$ assigns $x$ to $c_1$ with probability at least $1/2$, then the expected $\ell_1$ cost of the explainable clustering is at least $d/2$, while the optimal $\ell_1$ clustering cost is $d^{3/4}$ (by assigning $x$ to $c_2$).
>
> Case 2: If distribution $D$ assigns $x$ to $c_2$ with probability at least $1/2$, the expected $\ell_2$ cost of the explainable clustering is at least $d^{3/4}/2$, while the optimal $\ell_2$ clustering cost is $\sqrt{d}$ (by assigning $x$ to $c_1$).
>
> Q2. Additionally, regarding the dynamic clustering algorithm, normalization is applied to all centers. Will normalization also need to be applied to all nodes whenever new nodes are inserted? If so, how does this affect the efficiency of the algorithm?
>
> As far as your second question is concerned, the assumption that all future centers lie in $[-1,1]^d$ is used mainly for the ease of exposition. There is an implementation of the algorithm that does not depend on this assumption. Specifically, if the centers after request $t$ lie in $[-R,R]^d$, the weighted distribution $\tilde{f}(x)$ that we use to sample a threshold in $(y_{j-1}, y]$ does not depend on $R$ (we can compute the threshold by sampling a uniform random variable $U \in ((y_{j-1} - m)^p, (y-m)^p)$ and return $U^{1/p}$). Moreover, if all timestamps are multiplied by the same positive number $\alpha$, the analysis is not affected. Thus, in line 1171, we can sample $z\sim \exp(\lambda)$ with $\lambda = (y-m)^p - (y_{j-1} - m)^p$ (that does not depend on R), without affecting the analysis. We will add this clarification in the revised version of the paper.

---

> > ### Comment · Reviewer_tdu1 · 2025-08-05
> >
> > I agree with the answers. Thank you for the detailed examples.

---

### Official Review · Reviewer_3zna · 2025-06-30

**Clarity:** 3
**Significance:** 3
**Originality:** 3
**Rating:** 5
**Confidence:** 3

**Summary:**

Explainable clustering seeks to partition a set of points into clusters, where the cluster boundaries are given by coordinate cuts. One key question is how far the quality of clustering is with respect to an optimal clustering (which need not obey such restrictions on cluster boundaries). For the $k$-median objective, it is known that one can obtain $O(\log k)$-approximate explainable clustering and it is tight. The paper considers a more general objective where one needs to open $k$ centers (as in the $k$-median case), but the cost of assigning a client $x$ to a center $y$ is given by $||x-y||_p$ ($\ell_p$ norm). When $p=2$, $O(\log^{1.5} k)$-approximation algorithm was known (ignoring loglog factors). There are several results in the paper:

1. For the general $p$ norm objective, they give $O(p (\log k)^{1+1/p-1/p^2}$ approximation -- this improves the result for $p=2$ to $O(log^{1.25} k)$-approximation.
2. They show that the algorithm can be implemented in a dynamic setting where points are arriving online and the update time is $O(k polylog(n))$.

**Questions:**

1. Clearly distinguish the main itechnical deas from the [Makarychev and Shan (2021)] paper: in particular, is the new contribution the distribution with which to choose $\theta_t$ values?
2. In the dynamic setting, what are the new ideas...it seems the core algorithm for explainable clustering remains similar to prior work...so would the ideas for dynamic setting also extend to prior works?
3. What is the utility of these metrics in the context of explainable clustering?

**Ethical Concerns:**

["NO or VERY MINOR ethics concerns only"]

**Limitations:**

yes

**Quality:**

3

**Strengths And Weaknesses:**

Strengths:

1. Improved bounds for $p = 2$, and new algorithms for general $p$
2. The fact that these algorithms can be implemented in a dynamic setting is useful.
3. The problem has received lot of attention in the recent past, specially in the context of explainable AI
4. The techniques, though relying on prior work, still need fair bit of work for general $p$.

Weakness:
1. The template of the algorithm is very similar to prior work [Makarychev and Shan (2021)]
2. In general $p=1$ and $p=2$ are the most useful settings (and the $k$-means objective). So it is not clear how useful these results are.
3. An experimental validation showing that general $p$ matter in this context would have been useful.

---

> ### Author Rebuttal · Authors · 2025-07-29
>
> We thank the reviewer for constructive feedback.
>
> 1. As the reviewer correctly points out, our algorithm generalizes the one in Makarychev and Shan (2021), with the key difference being the modified distribution used to sample the threshold $\theta$. This change is essential for extending the guarantees to general $\ell_p$ norms. Beyond this algorithmic modification, our analysis introduces several new technical components. To achieve a tighter approximation ratio, we refine the analysis by partitioning cuts into three cases: (1) safe cut, (2) light cut, and (3) heavy and unsafe cut, while the previous work only considered light and heavy cuts. Additionally, the dynamic setting in this paper is entirely new. We will clarify these distinctions more explicitly in the revised version.
>
> 2. We thank the reviewer for the insightful comment and agree that the core algorithm for explainable clustering remains similar to prior work. The reviewer is also correct that the new ideas introduced for the dynamic setting can be extended to earlier algorithms for explainable k-medians under both the $\ell_1$ and $\ell_2$​ norms.
> While the high-level structure of the algorithm is preserved, the dynamic setting introduces several new technical challenges that require novel ideas:
>
> (1) We use exponential clocks to assign timestamps to decision nodes, allowing us to identify the earliest cut that separates an inserted center and update the threshold tree. This avoids recomputing the entire tree from scratch. Specifically, the exponential timestamps allow us to update the current tree efficiently in time O(d polylog k). Note that even classifying a single data point $x$ naively might require traversing a root-to-leaf path in O(k) time. Although exponential clocks were previously used in the analysis of $\ell_1$-based clustering algorithms (e.g., Gupta et al. 2023; Makarychev and Shan 2023), this is the first time they are used algorithmically for maintaining the dynamic explainable clustering.
>
> (2) Our static algorithm selects the anchor point as the coordinate-wise median of centers to control the number of centers separated by each cut. In the dynamic setting, the median of centers evolves as centers are inserted or deleted, which implicitly affects the threshold distribution. To address this, we keep track of the number of centers and rebuild the tree when necessary to maintain the anchor point as an approximate median of centers in each partition leaf call.
>
> 3. We thank the reviewer for the thoughtful question. Studying $\ell_p$ norm costs in clustering provides a flexible framework that interpolates between different clustering objectives, with practical and theoretical relevance. For instance, $\ell_1$ norm is known for its robustness to outliers, while $\ell_2$ norm is widely used due to its geometric interpretability and computational properties. Extending explainable clustering to general $\ell_p$​ norms allows practitioners to tailor the cost function to the needs of specific applications while preserving interpretability. Our goal is to understand how explainability impacts clustering quality across this broader family of objectives and to provide provable guarantees in this more general setting.

---

> > ### Comment · Reviewer_3zna · 2025-08-05
> >
> > Thanks. I am satisfied with the response.

---

### Official Review · Reviewer_ZVnr · 2025-07-02

**Clarity:** 3
**Significance:** 3
**Originality:** 3
**Rating:** 5
**Confidence:** 3

**Summary:**

This paper presents results for the explainable $k$-median problem for $\ell_p$ norm distance. The improve the state of the art for the case of $k$-means. The authors also extend their results to dynamic settings.

**Questions:**

What are the differences between the dynamic settings in this paper and the previous works?

**Ethical Concerns:**

["NO or VERY MINOR ethics concerns only"]

**Final Justification:**

Good paper with solid contribution.

**Limitations:**

yes

**Quality:**

3

**Strengths And Weaknesses:**

- The paper is well written and the ideas and techniques are explained clearly.

- The setting is interesting; although explainable algorithms provide lower quality clustering, they are interpretable and each decision on the tree depends on a single feature. In many real world applications, these are key constraints.

- The results are strong: i) The improvements for $\ell = 2$ is interesting, ii) For the higher values, this works presents the first result.

- The results in the dynamic setting are interesting but not as interesting as the offline setting. The setting seems to be different from multiple other k-median clustering problems in a dynamic setting. In most settings the points are inserted / deleted but in this work the centers are changing.

---

> ### Author Rebuttal · Authors · 2025-07-29
>
> We thank the reviewer for the detailed comments and appreciate the recognition of our results in the offline setting.
>
> We are not aware of any prior work on dynamic algorithms for explainable clustering. Dynamic algorithms have, however, been studied for unconstrained (non-explainable) $k$‑medians. While our dynamic algorithm is stated in terms of insertions and deletions of centers, it can be naturally combined with existing dynamic algorithms for unconstrained k-medians under $\ell_p$ norm to also handle dynamic updates to the data points themselves. Specifically, in settings where data points are inserted or deleted, we can maintain a good set of centers using a dynamic $k$-medians algorithm, and then use our algorithm to maintain the corresponding threshold tree for explainable clustering as these centers change. In particular, we can use a constant-factor approximation dynamic algorithm with small recourse that modifies only a small number of centers upon each insertion or deletion of a data point. We will include a clear discussion of this connection and state a formal corollary in the revised version.

---

> > ### Comment · Reviewer_ZVnr · 2025-08-04
> >
> > After reading the other reviews and the rebuttal I think the paper should be accepted and I keep my score.

---

### Official Review · Reviewer_X4Dr · 2025-07-06

**Clarity:** 3
**Significance:** 3
**Originality:** 4
**Rating:** 5
**Confidence:** 2

**Summary:**

The authors introduce an algorithm for explainable $k$-medians clustering problem under $\ell_p$ norm for generic $p \geq 1$. The authors also prove its approximation results, which were not known before. The author also discuss how this algorithm could be extended to dynamic settings.

**Questions:**

* Can the authors comment about the technical intuitions when extending the results from $p = 1,2$ to $p \geq 1$?
* Including a brief mention of hardness results in the introductory section would be appreciated, personally.

**Ethical Concerns:**

["NO or VERY MINOR ethics concerns only"]

**Final Justification:**

Solid results and should be accepted.

**Limitations:**

The paper does not include a paragraph addressing their limitations as it is a theory paper.

**Quality:**

4

**Strengths And Weaknesses:**

Strengths
* I highly appreciate the technical contributions and the elegance of the proofs presented in this paper.
* The authors also include a dynamic version of their algorithm to enhance their approach.

Weaknesses
* Practical discussions or experiments are not present; understandable as this is a theory paper.
* (minor point) In line 916 and 988, Hölder's ineqaulity should be used.

---

> ### Author Rebuttal · Authors · 2025-07-29
>
> We thank the reviewer for the detailed and constructive feedback.
>
> 1. The main intuition of extending the results from $p=1,2$ to general $p>=1$ is as follows. We build on the algorithmic framework of Makarychev and Shan (2021) for explainable k-medians under $\ell_2$ norm. The key modification lies in the sampling distribution for the threshold $\theta \in [0,R_t]$ in the Partition_Leaf function. Specifically, we sample theta according to the cumulative density function $x^p/(R_t)^p$, which corresponds to the density function $px^{p-1}/(R_t)^p$. This implies that for any $0<a<b<R_t$, the probability that the sampled threshold $\theta \in [a,b]$ is at most $p(b-a)b^p/(R_t)^p$, which is crucial for bounding the separation probability.  Our analysis is more refined than that of Makarychev and Shan (2021); as a result, it applies to all finite $p$ and yields improved theoretical guarantees for the $\ell_2$ norm.
>
> Note that these modifications are necessary to achieve a good approximation for $p \geq 1$. Specifically, for every value of $p$ we need to use a separate distribution. This is illustrated by the example we provide in our response to Reviewer tdu1 (see our answer to Q1 for Reviewer tdu1).
>
>
> 2. We have an example that shows an $\Omega(\log k)$ lower bound on the price of explainability for $k$‑medians under the $\ell_p$ norm for all $p \geq 1$. We will include it in the revised version of the paper. We will also expand the discussion of related lower bounds and hardness of approximation results, including the $\Omega(\log k)$ lower bound on the price of explainability for $k$‑medians under the $\ell_2$ norm due to Makarychev and Shan (2021) and the hardness results for explainable $k$‑medians under the $\ell_1$ norm by Gupta, Pittu, Svensson, and Yuan (2023).

---

> > ### Comment · Reviewer_X4Dr · 2025-08-06
> >
> > Thank you for the authors' detailed answer. I am satisfied with the rebuttal and will maintain my review with a positive score.

---

### Official Review · Reviewer_auTj · 2025-07-13

**Clarity:** 3
**Significance:** 3
**Originality:** 2
**Rating:** 5
**Confidence:** 3

**Summary:**

This paper addresses the problem of maintaining an explainable clustering under dynamic updates, specifically a decision tree clustering that approximates the $k$-medians objective. The goal is to construct and maintain a tree structure that enables interpretable clustering while achieving provable approximation guarantees. The authors present an algorithm that achieves an approximation factor of $O(p \log^{1 + 1/p - 1/p^2} k \log \log k)$ for the $k$-medians problem under the $\ell_p$ norm. Prior to this work, such guarantees were only known for the $\ell_1$ norm ($p = 1$), and for $p = 2$ (Euclidean case), the best-known explainable clustering algorithm had a weaker approximation factor of $O(\log^{3/2} k)$. A lower bound of $O(\log k)$ was already known. In addition to extending explainable clustering to general $\ell_p$ metrics, the authors also design a dynamic algorithm that maintains the same approximation factor under insertions and deletions.

**Questions:**

-

**Ethical Concerns:**

["NO or VERY MINOR ethics concerns only"]

**Final Justification:**

The results are solid and justify acceptance.

**Quality:**

3

**Strengths And Weaknesses:**

The $k$-median problem is both important and widely applicable, and designing a data structure that supports fast updates while maintaining clustering quality is a meaningful direction. While most practical applications involve $p = 1, 2$, extending the results to general $\ell_p$ norms is also of theoretical interest. Although the paper builds on several ideas from Makarychev and Shan (2021), the results are nontrivial and contribute meaningfully to the area, making the paper a reasonable candidate for acceptance.

---

> ### Author Rebuttal · Authors · 2025-07-29
>
> We thank the reviewer for the detailed and insightful comments.

---

### Decision · Program_Chairs · 2025-09-17

**Decision:**

Accept (spotlight)

**Comment:**

This paper studies explainable clustering in metric spaces under dynamic updates, the papers improves previous work on \ell_2 for the same problem and generalize to other \ell_p.

The studied problem is natural and of practical interest and the techniques in the paper generalize previous work. The paper would benefit from experiments and from a more detail comparison with Makarychev and Shan (2021) but overall it is above the acceptance bar.